# Hominoid SVA-lncRNA *AK057321* targets human-specific SVA retrotransposons in *SCN8A* and *CDK5RAP2* to initiate neuronal maturation

Monica J. S. Nadler[1,2], Weipang Chang[1,2], Ekim Ozkaynak[1,2], Yuda Huo[1,2,5], Yi Nong[1,2,5], Morgane Boillot[1,2], Mark Johnson[1,2], Antonio Moreno[1,2] & Matthew P. Anderson ◉ [1,2,3,4,5 ✉]

SINE-VNTR-Alu (SVA) retrotransposons arose and expanded in the genome of hominoid primates concurrent with the slowing of brain maturation. We report genes with intronic SVA transposons are enriched for neurodevelopmental disease and transcribed into long non-coding SVA-lncRNAs. Human-specific SVAs in microcephaly *CDK5RAP2* and epilepsy *SCN8A* gene introns repress their expression *via* transcription factor ZNF91 to delay neuronal maturation. Deleting the SVA in *CDK5RAP2* initiates multi-dimensional and in *SCN8A* selective sodium current neuronal maturation by upregulating these genes. SVA-lncRNA *AK057321* forms RNA:DNA heteroduplexes with the genomic SVAs and upregulates these genes to initiate neuronal maturation. SVA-lncRNA *AK057321* also promotes species-specific cortex and cerebellum-enriched expression upregulating human genes with intronic SVAs (e.g., *HTT*, *CHAF1B* and *KCNJ6*) but not mouse orthologs. The diversity of neuronal genes with intronic SVAs suggest this hominoid-specific SVA transposon-based gene regulatory mechanism may act at multiple steps to specialize and achieve neoteny of the human brain.

[1] Department of Neurology, Beth Israel Deaconess Medical Center, 330 Brookline Avenue, Boston, MA 02115, USA. [2] Department of Pathology, Beth Israel Deaconess Medical Center, 330 Brookline Avenue, Boston, MA 02115, USA. [3] Boston Children's Hospital Intellectual and Developmental Disabilities Research Center, 300 Longwood Avenue, Boston, MA 02115, USA. [4] Program in Neuroscience, Harvard Medical School, 300 Longwood Avenue, Boston, MA 02115, USA. [5] Present address: Neuroscience Therapeutic Focus Area, Regeneron, 777 Old Saw Mill River Road, Tarrytown, NY 10591, USA. ✉email: matthewanderson020@gmail.com

Advances in *Homo sapiens* cognitive ability compared to other primates occurred in parallel with changes in brain structure (e.g., expanded cortex and cerebellum) and function. The brain's core functional element, the glutamatergic neuron, displays neoteny or delayed maturation in human compared to other primates. Delayed neuronal maturation is considered a possible cellular basis of the extended brain growth that contributes to increased neuron numbers, size, and connectivity of the human brain[1–3]. Maturation of glutamatergic neurons from induced pluripotent stem cells (iPSCs) is delayed in human compared to its closest primate relatives, chimpanzee and bonobo[4]. Yet the molecular basis of this human neuronal neoteny is undefined.

SINE-VNTR-Alu (SVA) transposons are hominoid-primate-specific mobile genetic elements that transposed and became fixed in the genome during hominoid evolution[5,6]. As human-specific fixed transposons are considered a potential source of gene regulation that could underlie *Homo Sapiens* species divergence[7–9], we postulated SVAs, particularly those unique to human, might underlie the delayed neuronal maturation found in human.

Transposable elements are also enriched in expressed transcripts of long non-coding (lnc) RNAs[10], suggesting those containing SVA transposons might also have a role in hominoid-specific, transposon-based gene regulatory mechanisms.

We find that as human neurons mature, there is an increase of glutamatergic neuron-enriched SVA-lncRNA, *AK057321*. We increase *AK057321* expression to evaluate its effects on the expression of neurodevelopmental genes with intronic SVAs. These studies were performed in a human neuronal progenitor cell to assess the effects of these interventions on neuronal maturation. We also examine effects of intervening on KRAB-domain zinc finger transcription factor, ZNF91, that binds SVA sequences and recruits protein complexes to repress genes bearing an SVA[11,12]. To evaluate if intronic SVAs repress neurodevelopmental genes as a potential means to delay neuronal maturation, we deleted the gene's intronic SVAs. To complement these in vitro human neuronal progenitor cell studies, we also use humanized mice to examine whether SVA transposon-based gene regulation occurs in the brain in vivo. Finally, to begin to understand the molecular basis of this SVA transposon-based gene regulatory mechanism we evaluated for the possibility that complementary annealing of nucleic acid strands occurs between genomic and *AK057321* lncRNA SVA sequences. We also examine if the genomic SVA-binding transcription factor ZNF91[11,12] binds SVA-lncRNA *AK057321* as a potential mechanism of displacing the transcriptional repressor.

Altogether, these studies reveal that human-specific intronic SVA transposons together with SVA-binding transcription factor ZNF91 repress expression of human brain development genes (e.g., *CDK5RAP2* and *SCN8A*) to delay neuronal maturation. SVA sequences encoded in *AK0572321* of the SVA-lncRNA gene family, possibly through RNA:DNA heteroduplex formation and ZNF91 decoy-binding, release SVA-mediated gene repression to time the initiation of neuronal maturation.

## Results

### SVA-lncRNA *AK057321* expression increases during glutamatergic neuron maturation to upregulate expression of neurodevelopmental disease genes with intronic SVAs.

SVA transposons are unique to the great ape (hominoid) primate species lineage that includes the extant human, chimpanzee, gorilla, and orangutan (Fig. 1a)[5,6]. SVA-lncRNA *AK057321* (NONHSAT119982.2 in www.noncode.org), that is duplicated in rare cases of autism spectrum disorder (Supplementary Data 1), is found in human, chimpanzee, bonobo, and gorilla species and is encoded by three exons including a full-length SVA transposon sequence at the 3′ end (Fig. 1b, Supplementary Fig. 1, NONHSAT119982.2). SVA-lncRNA *AK057321* expression is strongest in brain and testis (Supplementary Fig. 2 and Supplementary Data 2; enriched in cortex and cerebellum, see *HTT* gene regulatory studies below; see also the expression of NONHSAT119982.2 in www.noncode.org). We observed that SVA-lncRNA *AK057321* expression increases 2-fold in NTera-2 progenitor cells as they are matured into neurons using *Ngn2* transcription factor and astrocyte co-culture (Fig. 1c and Supplementary Data 3) and 10-fold as pluripotent human iPSC cells differentiate into glutamatergic neurons (Supplementary Fig. 3). *Ngn2* and astrocyte co-culture promoted neuronal differentiation of NTera-2 progenitor cells as confirmed by the hyperpolarized resting membrane potential (Fig. 1d), the generation of sodium current-dependent action potential (Fig. 1d), and the visible neurite outgrowth (Fig. 1e). These results establish that SVA-lncRNA *AK057321* expression increases as neuronal progenitors mature into glutamatergic neurons and validate the feasibility of using the human NTera-2 pluripotent cell line for studies of neuronal maturation.

Using gene ontology term enrichment analysis, we found that human genes containing intronic SVAs are enriched for those involved in neurodevelopment and neurodevelopmental and neurological disease (Fig. 1f, Supplementary Data 4–6; enrichment for other biological function and disease gene ontology terms also seen). Functionally important genes underlying microcephaly, neuron dendrite and axon growth, synapse formation, and ion channels are found to contain intronic SVAs (Supplementary Figs. 4–7), with the SVA sequences frequently being human-specific, and sometimes displaying species-specific expansions (e.g., *ATRX*, *KDM6A*), tandem insertions (*CDK5RAP2*, *KDM6A*, *RAB2A*, *IL1RAPL1*, *KCNH1*) or even repeated integrations at multiple sites in the same gene (*KDM6A*, *ATP8A2*, *GRID1*, *NRXN1*). These findings are consistent with a bioinformatics study suggesting SVAs are associated with genes involved in central nervous system pathways[13].

Based on the SVA sequence complementarity in SVA-lncRNA *AK057321* and these intronic SVAs, we postulated SVA-lncRNA *AK057321* might regulate coding genes with intronic SVAs. To test this, we used a short hairpin (sh)RNA to deplete *AK057321* mRNA and measured expression of several genes with intronic SVAs in the displayed categories of interest. Depleting *AK057321* strongly downregulates microcephaly gene *CDK5RAP2* (human-specific tandem SVA_F, Supplementary Fig. 4) and epileptic encephalopathy gene *SCN8A* (human-specific SVA_D, Supplementary Fig. 7), as well as several other SVA-containing genes involved in neurogenesis, neuron excitability, synapse and dendrite formation, and mitochondrial function (Fig. 1g). By contrast, genes lacking SVAs, *HUNK* and *LSS*, were unaffected. Confirming the results with the shRNA, Cas9/sgRNA-mediated deletion of SVA-lncRNA *AK057321* (exon 3) reduces SVA-lncRNA *AK057321* expression while downregulating *CDK5RAP2*, *SCN8A*, and *CHAF1B* gene expression (Fig. 1h, i and Supplementary Fig. 8). These results establish that mimicking developmental increases of SVA-lncRNA *AK057321* promotes expression of neurodevelopmental genes with intronic SVAs.

Cas9 sgRNA-mediated deletion of SVA-lncRNA *AK057321* (exon 3) increases pluripotency marker gene expression (Fig. 1j; *POU5F1* and *NANOG* three days post transfection). These results suggest SVA-lncRNA *AK057321* promotes the switch from progenitor to neuron.

### Neuronal maturation is initiated by increasing SVA-lncRNA *AK057321* or deleting transcription factor *ZNF91* that binds

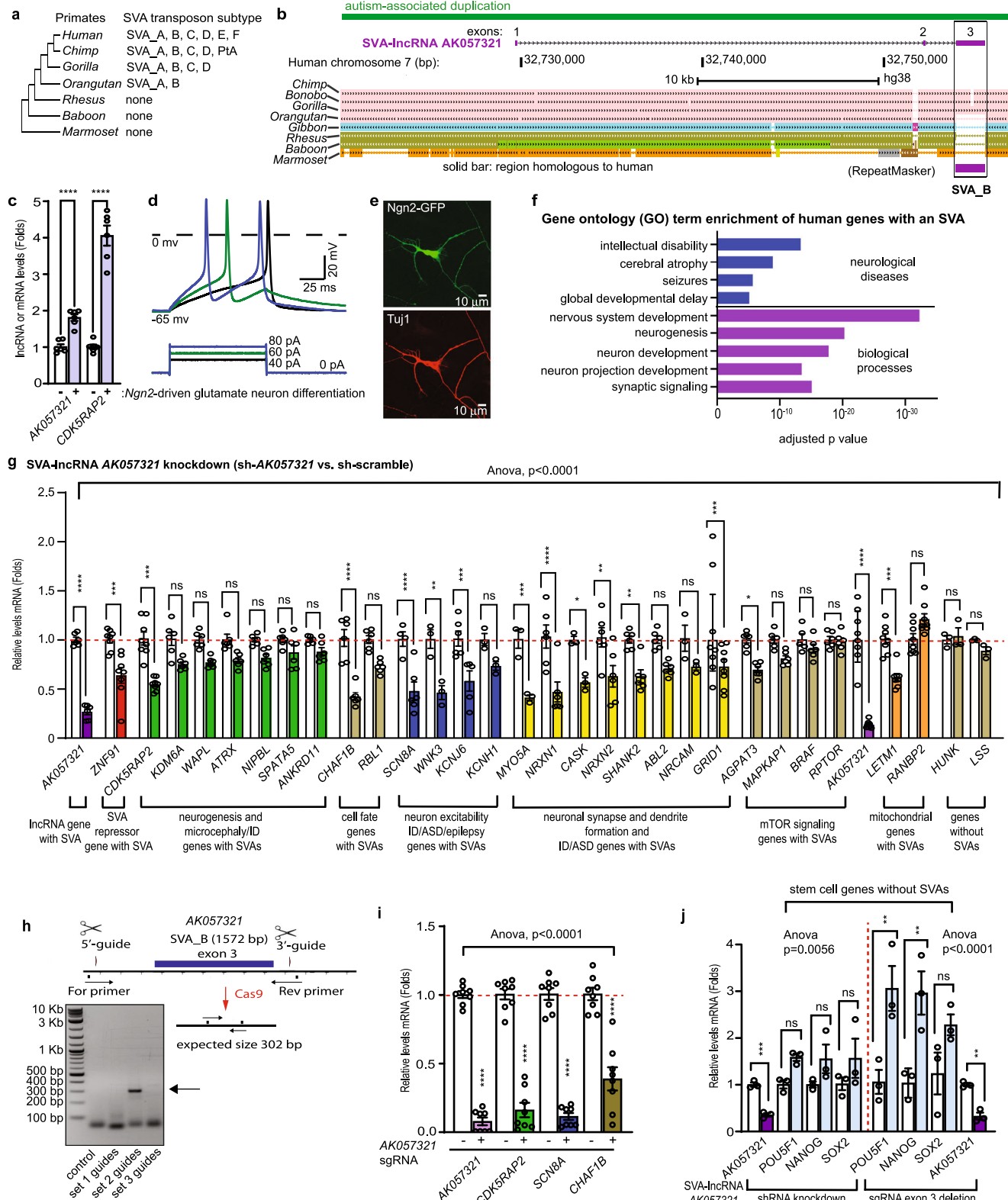

**SVAs to repress gene expression**. Based on our observations that SVA-lncRNA *AK057321* increases during neuronal maturation (Fig. 1c, Supplementary Fig. 3) and increases expression of neuronal genes with intronic SVAs such as the microcephaly gene *CDK5RAP2* (Fig. 1c, g–i), we reasoned SVA-lncRNA *AK057321* might function to initiate neuronal maturation by relieving the repression of genes by their intronic SVAs. For a second independent test of the potential role of SVAs in human neuronal

maturation, we also investigated the effects of manipulating *ZNF91*, a primate-specific Krüppel-associated box (KRAB) domain containing zinc finger protein (KZNF) that binds to intronic SVA sequences to repress gene expression[11,14,15]. Importantly, *ZNF91* is expressed in human neuronal progenitor NTera-2 cells[16]. Deleting *ZNF91* gene in human embryonic stem cells increases the expression of genes located near SVA sequences[12]. Prior luciferase reporter studies have shown *ZNF91*

**Fig. 1 SVA-lncRNA *AK057321* is upregulated during neuronal maturation, increases expression of neurodevelopmental disease genes with intronic SVAs, and reduces transcripts from stem/progenitor cell genes lacking SVAs. a** Schematic showing SVA subtypes in primate species. **b** Schematic of gene *AK057321*, a long non-coding RNA (lncRNA) expressing a full-length SVA (UCSC genome browser, GRCh38/gh38). **c–e** *Ngn2*-driven neuronal differentiation of NTera-2 cells increases SVA-lncRNA *AK057321* and *CDK5RAP2* expression (**c** plated on astrocytes six weeks, comparing to non-treated NTera-2 cells with *GAPDH* as the internal reference gene, unpaired two-tailed Student's *t* test, $n = 2$ biological replicates each measured in triplicate), generates Na$^+$ action potentials and hyperpolarizes membrane potential (**d**), and increases TuJ1 protein and neurite outgrowth (**e**, expressed GFP and neuron-specific Class III β-tubulin immunostaining). Unpaired two-tailed Student's *t* test. **f** Gene ontology (GO) analysis of human genes with intronic SVAs: biological process (purple bars) and disease (blue bars) term enrichment. **g** NTera-2 cells transfected with sh-*AK057321* or sh-scramble for 24 h were sorted and analyzed for gene expression by RT-qPCR. *HUNK* and *LSS* are controls without SVAs. One-way ANOVA ($F_{63, 298} = 11.37$, $p < 0.0001$; $n = 5$ to 6 biological replicate samples were pooled and measure using $n = 3$ technical replicates per gene) with Bonferroni post hoc tests. Data represent the mean ± SEM. **h** Schematic for Crispr/Cas9 SVA_B deletion strategy with DNA agarose gel image showing expected SVA-deletion PCR product size of 269 bp present only with set 2 guides which were used for all subsequent experiments. **i** NTera-2 cells transfected with Cas9/sgRNAs to delete the SVA in *AK057321* (vs. control, right) were sorted and analyzed for gene expression by RT-qPCR. *CDK5RAP2*, *SCN8A*, and *CHAF1B* expression are decreased. $n = 8$ biological replicates. One-way ANOVA ($F_{7,56} = 92.36$, $p < 0.0001$) with Bonferroni's post hoc tests. **j** NTera-2 cells were transfected with sh-*AK057321* or sh-scramble (left) or NTera-2 cells were transfected with Cas9/sgRNAs to delete the SVA in *AK057321* (vs. control, right). After three days, cells were sorted and analyzed for gene expression by RT-qPCR with *GAPDH* as the internal reference gene. $n = 3$ biological replicates. Unpaired Student's *t* test. One-way ANOVA for sh-*AK057321* vs sh-scramble ($F_{5,12} = 6.809$, $p = 0.0031$) and one-way ANOVA for Cas9/sgRNAs ($F_{7,16} = 10.24$, $p < 0.0001$. ****$p < 0.0001$, ***$p < 0.001$, **$p < 0.01$, *$p < 0.05$, ns $p > 0.05$ versus control.

binds SVAs to repress gene expression[11,16]. We predicted deletion of *ZNF91* would increase expression of genes containing intronic SVA sequences in human NTera-2 progenitor cells.

We compared the effects of deleting *ZNF91* to those of increasing SVA-lncRNA *AK057321*, each intervention hypothesized to increase expression of genes repressed by ZNF91 repressor complex binding to intronic SVA sequences (Fig. 2a). Over-expressing SVA-lncRNA *AK057321* or deleting *ZNF91*, each independently, increases *CDK5RAP2* gene expression (Fig. 2b, c), a microcephaly gene with an intronic human-specific SVA_F that increases during neuronal maturation (Fig. 1c).

To quantify neuronal maturation, we measure neurophysiologic (sodium and potassium currents, resting membrane potential, membrane capacitance, and action potential firing) and morphologic (cell size, shape and dendrite formation) features of the maturing neuron[17–19]. We found human NTera-2 progenitor cells undergo neuronal maturation in response to transcription factor *Ngn2* when co-cultured with astrocytes (Fig. 1c–e). We observed that increasing SVA-lncRNA *AK057321* expression or deleting *ZNF91* gene in human NTera-2 progenitor cells co-cultured with astrocytes (but without added *Ngn2*) initiates early stages of neuronal maturation. Both interventions increase inward voltage-gated sodium current (Fig. 2d) and outward potassium current (Supplementary Fig. 9). The sodium current is inhibited by voltage-gated sodium channel blocker tetrodotoxin (Fig. 2e). These interventions also hyperpolarize resting membrane potential (RMP; Fig. 2f, g) and increase membrane capacitance (Fig. 2h, i). Consistent with the increased membrane capacitance reflecting increased membrane area, we observed neurite outgrowth, a morphologic property of maturing neurons, by confocal microscopy in cells treated with SVA-lncRNA *AK057321* (Fig. 2j, k). Quantification of neuronal morphology (Sholl analysis) confirmed SVA-lncRNA *AK057321* overexpression increases process length (Fig. 2l), cell area (Fig. 2m), and average mean and maximum process length (Fig. 2n, p) while decreasing circularity (Fig. 2o). SVA-lncRNA *AK057321* also increases average mean length and minimum length of processes, and average process and tip number, while increasing microtubule-associated protein 2 (MAP2), a marker of neuronal differentiation (Supplementary Fig. 10). The results indicate increasing SVA-lncRNA *AK057321* or depleting ZNF91 increases *CDK5RAP2* expression while initiating functional and structural neuron maturation.

**SVA-lncRNA *AK057321* initiates neuronal maturation by increasing *CDK5RAP2* expression otherwise repressed by an intronic tandem SVA_F.** We reasoned that removing specific genomic SVAs might identify the pathway whereby SVA-lncRNA *AK057321* drives neuronal differentiation. We first focused on *CDK5RAP2* because of its major role in human microcephaly and requirement during neurogenesis[20–26], its human-specific tandem SVA_F (Fig. 3a), and its strong downregulation with *AK057321* loss (Fig. 1g–i). *CDK5RAP2* expression was also upregulated by multiple interventions that promote neuronal maturation [increased expression of *Ngn2* (Fig. 1c), increased SVA-lncRNA *AK057321* expression (Fig. 2b) or decreased *ZNF91* expression (Fig. 2c)].

Using the CRISPR/Cas9-sgRNA approach, we deleted the tandem SVA_F in *CDK5RAP2* in NTera-2 cells (Fig. 3a, b and Supplementary Fig. 11). Loss of the intronic SVA sequences increases *CDK5RAP2* expression (Fig. 3c), underscoring the inhibitory effect of this tandem SVA_F on *CDK5RAP2* expression. The intervention also decreases stem/progenitor cell gene expression (*NANOG* and *POU5F1*; Fig. 3c), suggesting deletion of this 2.4 kb intronic sequence might suffice to initiate neuronal maturation.

As an additional test of the role of *CDK5RAP2* expression on neuronal maturation, we used an shRNA to decrease, and a cDNA to increase *CDK5RAP2* expression (RT-qPCR, Fig. 3d, e). Because *CDK5RAP2* overexpression is toxic[26], *CDK5RAP2* cDNA was expressed from a Tet-On controlled promoter (RT-qPCR, Fig. 3e).

As observed previously (Fig. 2d, e), and repeated experimentally here, increasing SVA-lncRNA *AK057321* increases inward voltage-gated sodium currents (Fig. 3f, g). Given that SVA-lncRNA *AK057321* increases *CDK5RAP2* expression (Fig. 2b), we blocked this increase with the *CDK5RAP2* shRNA (Fig. 3d, f–i). *CDK5RAP2* shRNA blocked *AK057321*-induced increases of sodium currents (Fig. 3f, g). Moreover, over-expressing *CDK5RAP2* itself increases sodium currents (Fig. 3f, g). Most importantly, deleting just the tandem SVA_F sequences in *CDK5RAP2* increases sodium currents (Fig. 3f, g). As observed in response to increases of SVA-lncRNA *AK057321* or decreases of *ZNF91* expression, the interventions that increase *CDK5RAP2* expression (overexpression and SVA deletion) also hyperpolarize resting membrane potential (Fig. 3h) and increase membrane capacitance (Fig. 3i). *CDK5RAP2* shRNA blocks the SVA-lncRNA *AK057321*-induced hyperpolarization (Fig. 3h) and increase of membrane capacitance (Fig. 3i). These results show the human-specific intronic tandem SVA_F in *CKD5RAP2* represses its

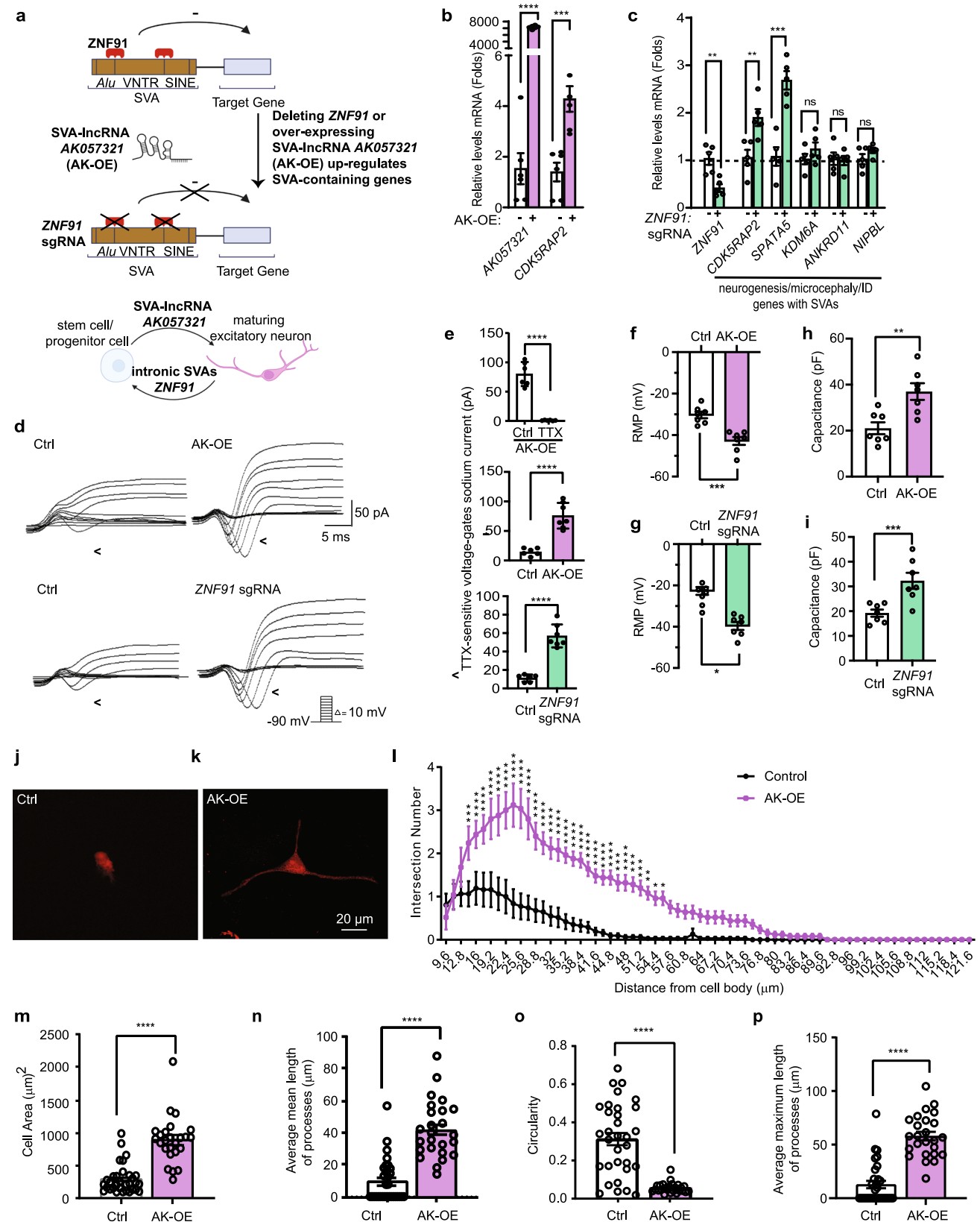

expression to block neuronal maturation and SVA-lncRNA *AK057321* releases this block by derepressing *CDK5RAP2*.

**Deleting the human-specific SVA_D in *SCN8A* selectively increases voltage-gated sodium current**. We reasoned that intronic SVAs in other genes regulated by SVA-lncRNA *AK057321* (Fig. 1g) might delay specific aspects of neuronal maturation and chose to examine *SCN8A* because it encodes the pore forming α subunit of a tetrodotoxin-sensitive, voltage-gated sodium channel and is needed to generate action potentials during early neuronal maturation[27]. *SCN8A's* role in early

**Fig. 2 Neuronal maturation is initiated by increasing SVA-lncRNA _AK057321_ or decreasing the _ZNF91_ transcription factor that binds SVAs to repress genes. a** schematic illustrates ZNF91 transcription factor binds to the _Alu_-VNTR and VNTR-SINE borders within SVAs[12] and inhibits nearby gene expression[11,12,16]. When SVA-lncRNA _AK057321_ expression is increased, or _ZNF91_ is deleted, SVA-containing genes are upregulated, and stem/progenitor cells mature into neurons (Created with Biorender.com). **b** _AK057321_ and _CDK5RAP2_ expression in NTera-2 cells transfected with either SVA-lncRNA _AK057321_ cDNA (OE) or control plasmid for 24 h (_GAPDH_, reference; $n = 6$ biological replicates). Unpaired Student's _t_ test. **c**, Gene regulation by Cas9/sgRNA deletion of _ZNF91_ in NTera-2 cells with _GAPDH_ as the internal reference gene. $n = 5$ biological replicates. Unpaired Student's _t_ test. **d** Representative inward current traces in _ZNF91_ sgRNA and _AK057321_ OE treatment groups in response to step depolarization ($-90$ mV to 0 mV). Inward currents were recorded in voltage clamp mode. **e** Tetrodotoxin (TTX) sensitive inward currents (control vs. TTX) were significantly larger in _AK057321_ OE and _ZNF91_ sgRNA and treatment groups than that in the control group. **f, g** Resting membrane potentials (RMP) is hyperpolarized in _AK057321_ OE ($n = 7$) and _ZNF91_ sgRNA ($n = 7$) compared to control. **h, i** Cell capacitance is increased in _AK057321_ OE ($n = 7$) and _ZNF91_ sgRNA ($n = 7$) compared to control. **j, k** Representative images of NTera-2 cell morphology with or without _AK057321_ overexpression 15 days after plating. Scale bar represents 20 μm. **l** Sholl analysis on the processes of Ntera-2 cells with or without _AK057321_ OE. **m** Quantification of cell area from control and _AK057321_ OE in NTera-2 cells. **n** Quantification of average mean length of processes. **o** Quantification of circularity based on cell morphology. **p** Quantification of average maximum length of processes. All graphs are presented as Mean ± SEM. Control, $N = 31$, AK overexpression, $N = 25$. In (**l**), ****$p < 0.0001$, ***$p < 0.001$, **$p < 0.01$, *$p < 0.05$ two-way ANOVA with Sidak's post hoc test. **b, c, e–i, m–p** Unpaired two-tailed Student's _t_ test. Data represent the mean ± SEM.****$p < 0.0001$, ***$p < 0.001$, **$p < 0.01$, *$p < 0.05$, ns $p > 0.05$ versus control.

neuronal excitability is evidenced by the ability of gain-of-function mutations to cause early infantile epileptic encephalopathy and loss-of-function mutations to cause intellectual disability[28–30]. Increasing and decreasing SVA-lncRNA _AK057321_ was found to upregulate and down-regulate, respectively, _SCN8A_ expression (Supplementary Fig. 12a, b, Fig. 1g–i). Moreover, CRISPR/Cas9-sgRNA-mediated deletion of _ZNF91_ upregulates _SCN8A_ expression (Supplementary Fig. 12c). To determine if the human-specific intronic SVA in _SCN8A_ impacts any aspects of neuronal maturation as observed for _CDK5RAP2_, we used CRISPR/Cas9-sgRNA to delete this SVA in NTera-2 cells co-cultured with astrocytes (Fig. 4a and Supplementary Figs. 12–13). We analyzed TTX-sensitive sodium currents in voltage clamp mode and representative traces are shown (Fig. 4b). Deleting the SVA in _SCN8A_ increases sodium currents (Fig. 4c) without altering resting membrane potential (Fig. 4d) or capacitance (Fig. 4e). The results suggest the human-specific intronic SVA_D in _SCN8A_ selectively delays maturation of voltage-gated sodium channel currents.

We compared the time course of neuronal maturation across different interventions to understand their kinetics and to evaluate if these parameters change spontaneously in control cells. SVA-lncRNA _AK057321_ overexpression, _ZNF91_ depletion, _SCN8A_ SVA deletion, _CDK5RAP2_ overexpression, or _CDK5RAP2_ SVA deletion (but not control or _CHAF1B_ SVA deletion, see also below) increase sodium spike amplitude over the 15-day time period (Fig. 4f, g and Supplementary Fig. 12d, e). Increases of inward voltage-sensitive sodium currents displayed a similar time course across all interventions but the increase is absent in control and with _CHAF1B_ SVA deletion (Fig. 4h). Importantly, unlike the other interventions, _SCN8A_ SVA deletion did not hyperpolarize resting membrane potential or increase membrane capacitance at any time point, consistent with its specialized role in regulating voltage-gated sodium currents (Fig. 4i, j).

**Deleting the SVA_B in _CHAF1B_ increases its expression and promotes proliferation and stem cell gene expression.** Some intronic SVAs may control functions beyond neuronal maturation. We examined functions of the SVA in chromatin assembly factor 1B (_CHAF1B_) because of its intronic SVA_B (Supplementary Fig. 14a) and strong downregulation when SVA-lncRNA _AK057321_ is depleted (Fig. 1g–i). CHAF1B is a component of a complex required for DNA replication and chromatin formation[31–33] that promotes proliferation[33] and maintains cell identity[34]. Deleting SVA-repressive _ZNF91_, or increasing SVA-lncRNA _AK057321_ upregulates _CHAF1B_ expression (Supplementary Fig. 14b). Deleting the SVA_B in _CHAF1B_ (CRISPR/

Cas9-gRNA technique, Supplementary Fig. 14c and Supplementary Fig. 15) also increases _CHAF1B_ expression and increases expression of pluripotency-promoting genes (_POU5F1_, _NANOG_, and _SOX2_) while reducing expression of neuron-differentiation promoting transcription factor _FOXG1_ (Supplementary Fig. 14d). Deleting the _CHAF1B_ SVA also fails to increase sodium spikes (Fig. 4f, 15 days) or currents (Supplementary Fig. 14e, f), to hyperpolarize resting membrane potential (Supplementary Fig. 14g), or to increase membrane capacitance (Supplementary Fig. 14h). Interestingly, deleting the SVA in _CHAF1B_ (but not in _CDK5RAP2_ or _SCN8A_) also increases the number of cells at 3 and 5 days following the intervention (Supplementary Fig. 14i–n). These results suggest the intronic SVA in _CHAF1B_ may repress its expression to inhibit proliferation of a stem cell pool generated by human NTera-2 progenitor cells.

**Human-specific cortex and cerebellum-enriched expression of Huntingtin's disease gene _HTT_ with an intronic SVA is driven by SVA-lncRNA _AK057321_.** _HTT_ mRNA is expressed in the developing and adult brain in both mice and humans[35,36] and contains an intronic SVA_B shared with chimpanzee and gorilla (Fig. 5a, length expanded in human and chimpanzee). Dominantly inherited mutations of _huntingtin_ (_HTT_) that expand a polyglutamine tract in exon 1 of the HTT protein underlie the neurological disorder Huntingtin's disease (HD)[37] with degeneration of medium spiny neurons of the striatum, motor deficits, and impaired cognitive executive functioning[37]. To investigate if SVA-lncRNA _AK057321_ not only acts during neuronal maturation but also in the fully mature brain to regulate expression of genes with an intronic SVA, we generated transgenic mice with a full-length human SVA-lncRNA _AK057321_ gene [derived from a bacterial artificial chromosome (BAC) vector human gene library, corresponding to the region duplicated in autism[38] (Fig. 1b, Supplementary Data 1, and Supplementary Fig. 2)]. We then crossed these SVA-lncRNA _AK057321_ transgenic mice to mice containing the human _HTT_ transgene (carried in yeast artificial chromosome vector and bearing an HD model mutation, YAC128 mice[39]).

In human brain tissue, we found SVA-lncRNA _AK057321_ and _HTT_ mRNA each display a 1.5 to 2-fold enriched expression in human cortex and cerebellum (Cereb) relative to brainstem (Bs) (Fig. 5b, c). This cortex and cerebellum-enriched expression of SVA-lncRNA _AK057321_ in human brain is reconstituted in _AK057321_ transgenic mice (Supplementary Fig. 16). Mouse brains expressing both SVA-lncRNA _AK057321_ and human _HTT_ transgenes also reconstitute the human-specific cortex and cerebellum-enriched expression pattern of _HTT_ mRNA (Fig. 5d). Whereas in mice expressing human _HTT_ alone, the _HTT_ mRNA

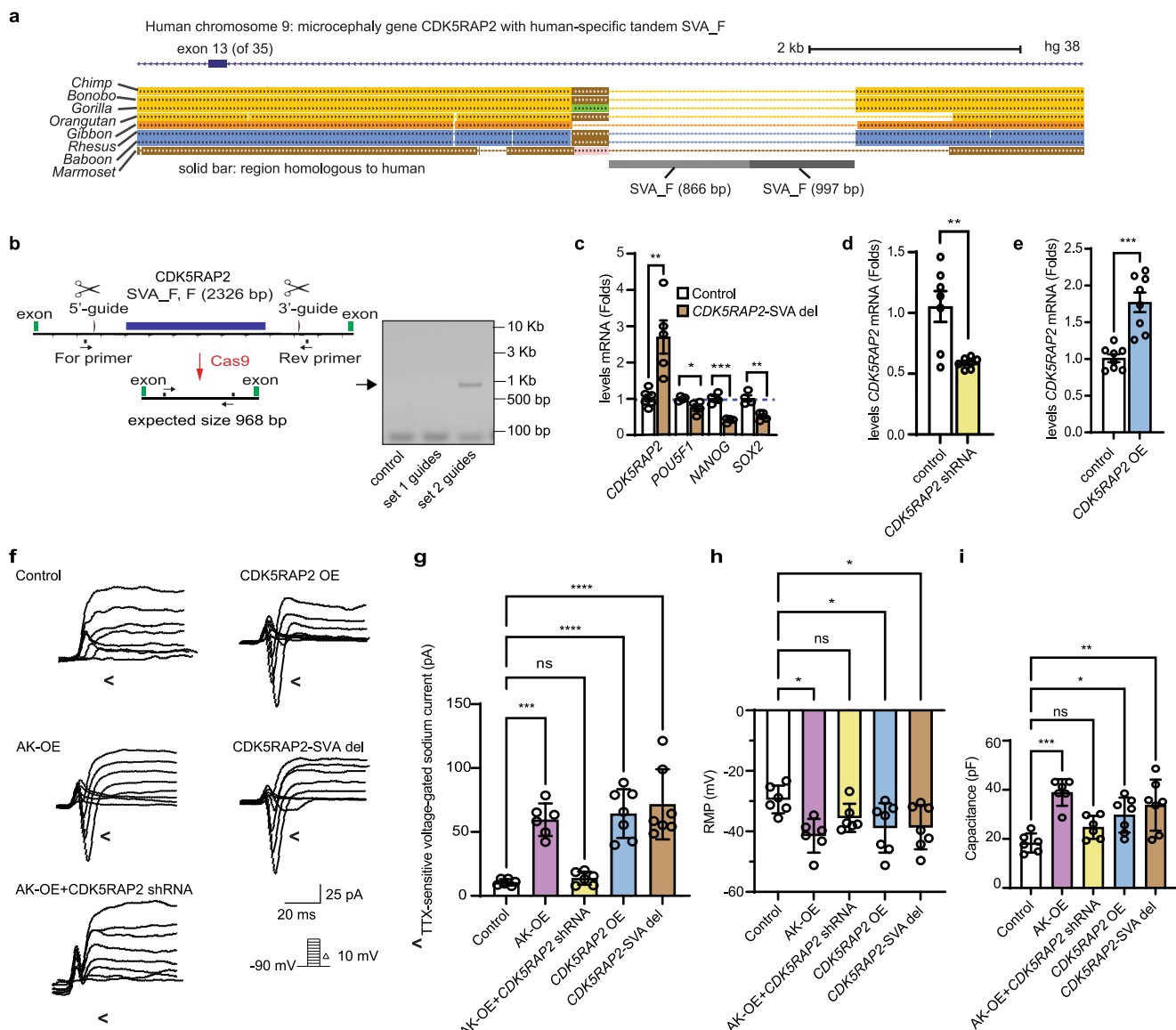

**Fig. 3 A human-specific tandem SVA-F in *CDK5RAP2* represses its expression to delay, whereas SVA-lncRNA *AK057321* enhances *CDK5RAP2* expression to promote neuronal maturation. a** Tandem human-specific SVA_F in microcephaly gene *CDK5RAP2* gene intron (UCSC genome browser, GRCh38/gh38). **b** Schematic for Crispr/Cas9 tandem SVA_F, F deletion strategy with DNA agarose gel image showing expected SVA-deletion PCR product size of 947 bp present only in set 2 guides which were used for all subsequent experiments. **c**, NTera-2 cells transfected with set 2 guides were sorted and analyzed for gene expression by RT-qPCR. $n = 4$ biological replicates. **d** RT-qPCR of NTera-2 cells transfected with *CDK5RAP2* shRNA show decreased *CDK5RAP2* mRNA expression compared to control cells. $n = 7$ biological replicates. **e** RT-qPCR from NTera-2 cells transfected with *CDK5RAP2* cDNA and induced with doxycycline have increased *CDK5RAP2* gene expression. $n = 8$ biological replicates. **c–e** Unpaired two-tailed Student's *t* test for each with *GAPDH* as the internal reference gene. **f** TTX-sensitive currents were recorded in voltage clamp mode. Representative sodium current traces in response to step depolarization were shown. **g** TTX-sensitive inward currents were larger in *AK057321* OE ($n = 7$), *CDK5RAP2*-OE ($n = 7$) and *CDK5RAP2*-SVA del ($n = 6$) del groups than those in the control group. *AK057321* OE + *CDK5RAP2* shRNA has no significant difference compare to control. One-way ANOVA ($F_{5,36} = 29.44$, $p < 0.0001$) with Bonferroni post hoc tests. **h** Resting membrane potential (RMP) is hyperpolarized in *AK057321* OE ($n = 7$), *CDK5RAP2*-OE ($n = 7$) and *CDK5RAP2*-SVA del ($n = 6$) del groups than those in the control group. *AK057321* OE + *CDK5RAP2* shRNA has no significant difference compared to control. One-way ANOVA ($F_{5,36} = 29.44$, $p < 0.0001$) with Bonferroni post hoc tests. **i** Capacitance was larger in *AK057321* OE ($n = 7$), *CDK5RAP2*-OE ($n = 7$) and *CDK5RAP2*-SVA del ($n = 6$) del groups than those in the control group. *AK057321* OE + *CDK5RAP2* shRNA has no significant difference compared to control. One-way ANOVA ($F_{5,36} = 29.44$, $p < 0.0001$) with Bonferroni post hoc tests. Data represent the mean ± SEM. ****$p < 0.0001$, ***$p < 0.001$, **$p < 0.01$, *$p < 0.05$, ns $p > 0.05$.

transcript lacks cortex and cerebellum-enriched expression being equal across brain regions (Fig. 5d), as observed for native mouse *Htt* mRNA (Fig. 5e). Thus, SVA-lncRNA *AK057321* drives a cortex and cerebellum-enriched expression pattern of human *HTT* (Fig. 5d) without altering expression of mouse *Htt* that lacks the intronic SVA (Fig. 5e). The results suggest SVA-lncRNA *AK057321*

may confer cortex and cerebellum-enriched of multiple genes contain an intronic SVA (Fig. 1f, g and Supplementary Figs. 4–7).

**SVA-lncRNA *AK057321* also upregulates expression of human Chromosome 21 genes with intronic SVAs in the brain.** Effects

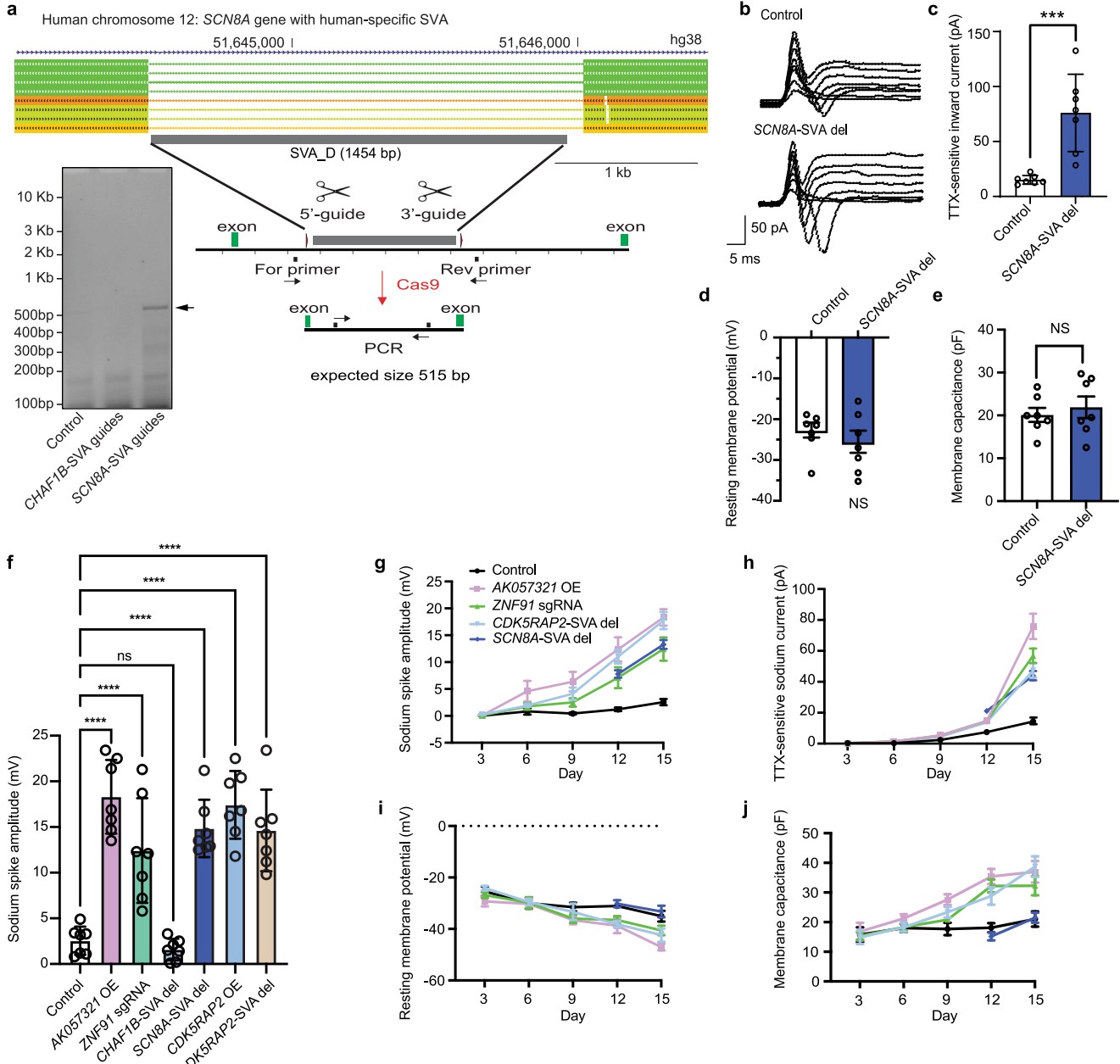

**Fig. 4 A human-specific SVA-D in *SCN8A* delays maturational increases of voltage-gated sodium current. a** Schematic showing SVA_D located within the *SCN8A* gene is a human-specific insertion (UCSC genome browser, GRCh38/gh38) and strategy for Crispr/Cas9 SVA_D deletion strategy with DNA agarose gel image showing expected SVA-deletion PCR product size of 515 bp present only with transfection of *SCN8A* guides which were used for all subsequent experiments. **b** Representative sodium current traces in SCN8A-SVA del group in response to step depolarization were shown. **c** TTX-sensitive current in *SCN8A*-SVA del group (n = 7) is higher than in control (n = 7). Unpaired Student's *t* test. ***p < 0.001. **d** Resting membrane potential (RMP) in *SCN8A*-SVA del (n = 7) and control (n = 7) group has no significant difference. Unpaired Student's *t* test. **e**, Capacitance in *SCN8A*-SVA del (n = 7) and control (n = 7) group has no significant difference. Unpaired Student's *t* test. Whole-cell patch-clamp recording of NTera-2 cell up to day 15 in culture in different treatment groups: **f** Current injection (300 pA) at day 15. **g** Change in voltage (Delta V). **h** TTX-sensitive inward currents. **i** Resting membrane potential. **j** Cell membrane capacitance. Two-way ANOVA with Bonferroni post hoc tests. Data represent the mean ± SEM. ****p < 0.0001, ***p < 0.001, ns p > 0.05.

of SVA-lncRNA *AK057321* on genes with intronic SVAs on human chromosome 21 were also examined in the mouse brain. We crossed SVA-lncRNA *AK057321* transgenic mice to mice with one copy of human chromosome 21, a transchromosomic model of the human intellectual disability disorder Down Syndrome[40]. SVA transposons on human chromosome 21 (Chr21) were identified using Repeat Masker in the UCSC genome browser (GRCh38/hg38). SVA-lncRNA *AK057321*

upregulates the expression of 3 of 7 genes with an intronic SVA in mouse cortex including *POFUT2*, *CHAF1B*, and *KCNJ6* (Fig. 6a, b). *AK057321* did not regulate mouse orthologs (Fig. 6c) or genes on human Chr21 nearby but lacking SVAs (Supplementary Fig. 17). Other human genes on Chr21 lacking intronic SVAs were also unaffected (Fig. 6d). *AK057321* also increases *CHAF1B* expression in cortex of mice with a full-length human *CHAF1B* transgene (Fig. 6e). These data provide further examples where

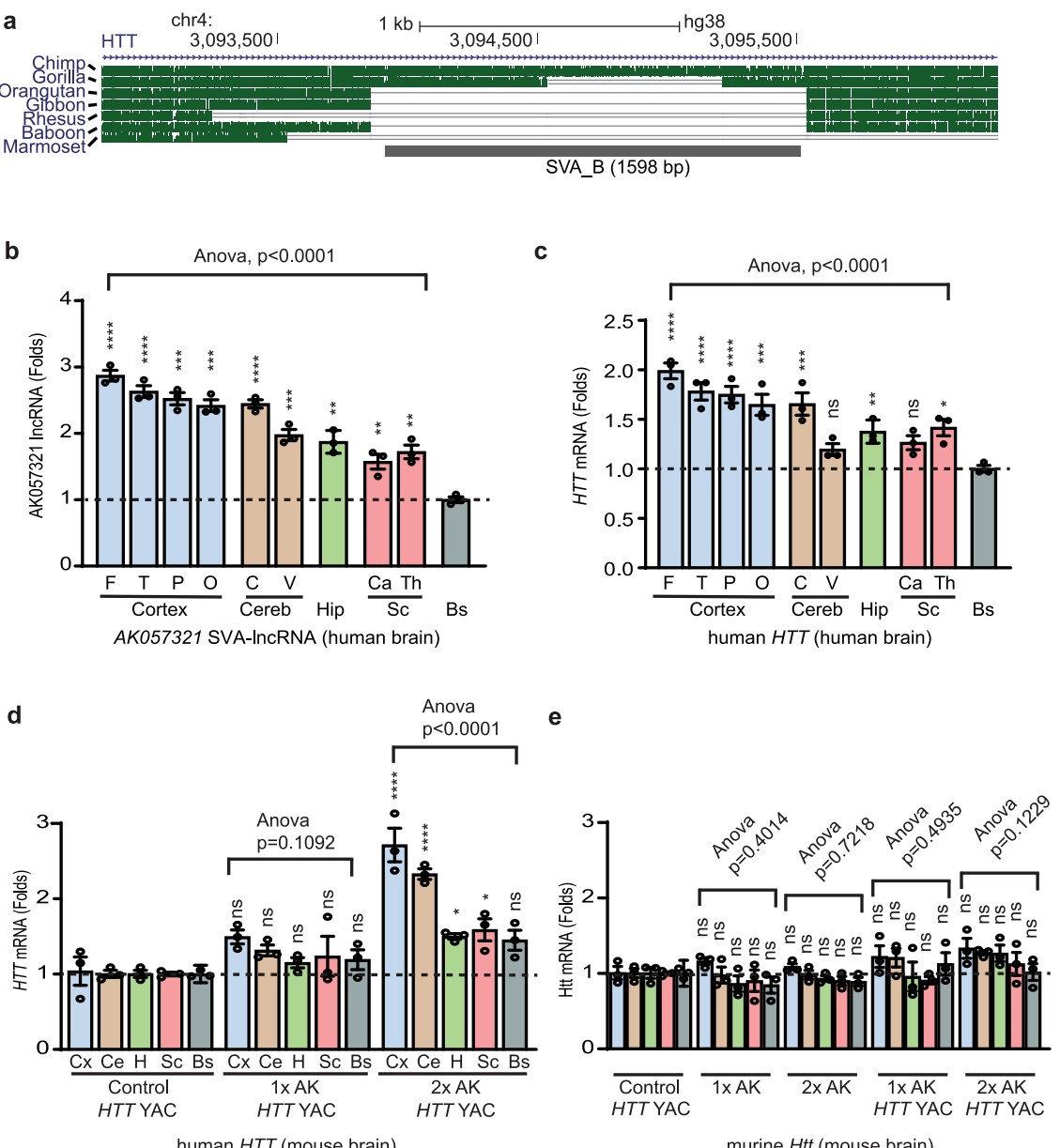

**Fig. 5 SVA-lncRNA *AK057321* reconstitutes human-specific cerebral cortex and cerebellum-enriched expression of human *HTT* gene in transgenic mice. a** Illustration of SVA_B retrotransposon located within the *HTT* gene, UCSC genome browser (GRCh38/gh38). **b** Native SVA-lncRNA *AK057321* transcript levels differ across human brain regions compared to levels in Bs. One-way ANOVA ($F_{9,20} = 43.19$, versus control Bs). **c** *HTT* transcript analysis with quantitative RT-qPCR on human brain tissue shows *HTT* transcript expression varies across human brain regions, compared to levels in Bs. One-way ANOVA ($F_{9,20} = 14.20$, $p < 0.0001$) with Bonferroni tests. **d** *HTT* is increased more in the cortex and cerebellum compared to other brain regions in 2x *AK057321/HTT* mice. One-way ANOVA ($F_{9,20} = 15.77$, $p < 0.0001$) with Bonferroni tests. No difference is seen in 1x *AK057321/HTT* and *HTT* control mice. One-way ANOVA ($F_{9,20} = 1.934$, $p = 0.1052$) with Bonferroni tests. **e** Mouse *Htt* transcript across brain regions. One-way ANOVA with Bonferroni tests for 2x *AK057321/HTT* ($F_{9,20} = 1.840$, $p = 0.1229$); for 1x *AK057321/HTT*, ($F_{9,20} = 0.9679$, $p = 0.4935$); for 2x *AK057321* ($F_{9,20} = 0.6754$, $p = 0.7218$); and for 1x *AK057321* ($F_{9,20} = 1.107$, $p = 0.4014$). **c**–**e** *RPL13A* was used as the internal reference gene. For all experiments above, 6 mice per genotype were used. qRT-PCR performed on $n = 3$ RNA samples (each pooled from two mice). Data represent the mean ± SEM. ****$p < 0.0001$, ***$p < 0.001$, **$p < 0.01$, *$p < 0.05$, ns $p > 0.05$. AK: *AK057321*, F: frontal cortex, T: temporal cortex, P: parietal cortex, O: occipital cortex, Cx: cortex, V: vermis, H/Hip: hippocampus, Cereb/Ce: cerebellum, C: cerebellar cortex, V: cerebellar vermis, Sc: subcortex, Ca: caudate, Th: thalamus, Bs: brainstem.

SVA-lncRNA *AK057321* upregulates expression of human genes carrying but not mouse orthologs lacking an intronic SVA in the brain.

**SVA-containing gene transcripts are increased in human compared to mouse cerebellum.** Based on the finding that SVA-lncRNA *AK057321* drives a cerebellum-enriched expression of SVA-containing human *HTT* but not mouse *Htt* lacking an SVA,

we compared the expression of several genes with intronic SVAs in human and mouse cerebellum samples. We designed RT-qPCR primers to conserved sequences in human and mouse for genes containing intronic SVAs and identified five genes *CHAF1B*, *IARS*, *PI4K2B*, *OSBP*, and *FNIP2* where RT-qPCR primers amplified a product in cerebellum of both species. Transcripts for each gene were increased in human compared to mouse in cerebellum tissues (Supplementary Fig. 18a). Expression of control

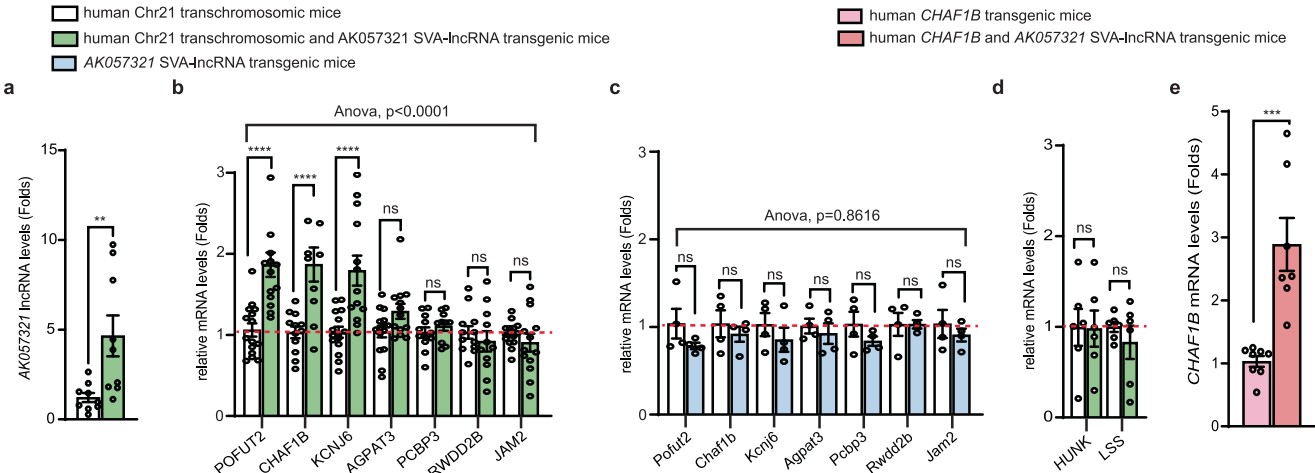

**Fig. 6 SVA-lncRNA *AK057321* increases expression of *CHAF1B, KCNJ6,* and *POFUT2* genes with intronic SVAs on human chromosome 21 but not mouse orthologs in mouse cortex. a** RT-qPCR of *AK057321* transcript levels shown in mouse cortex from human Chr21 transchromosomic/1x *AK057321* mice compared to human Chr21 transchromosomic mice. $n = 9$ biological replicates. Unpaired Student's $t$ test. **b** RT-qPCR expression levels of human genes with intronic SVAs from mouse cortex of human Chr21 transchromosomic/1x *AK057321* mice compared to human Chr21 transchromosomic mice. $n = 11$–14 biological replicates. One-way ANOVA ($F_{13,165} = 10.38$, $p < 0.0001$) with Bonferroni post hoc tests. **c** RT-qPCR of ortholog mouse gene expression from mouse cortex in *AK057321* SVA-lncRNA transgenic mice compared to wild-type mice. $n = 4$ biological replicates. One-way ANOVA ($F_{13,42} = 0.5728$, $p = 0.8616$) with Bonferroni post hoc tests. **d** Expression levels of two human Chr21 genes without intronic SVAs shown from mouse cortex of human Chr21 transchromosomic/1x *AK057321* mice compared to human Chr21 transchromosomic mice. Measured by RT-qPCR. $n = 6$ biological replicates. Unpaired Student's $t$ test. **e** Expression level of *CHAF1B* from mouse cortex of human *CHAF1B* transgenic mice compared to human *CHAF1B* transgenic/*AK057321* transgenic mice. Measured by RT-qPCR. $n = 7$ biological replicates. Unpaired Student's $t$ test. **a–e** B2M used as the internal reference gene. Data represent the mean ± SEM. ****$p < 0.0001$, ***$p < 0.001$, **$p < 0.01$, ns $p > 0.05$.

genes for gray matter (*PSD95*) and white matter (*APOD*) were equivalent suggesting similar tissue composition and RNA sampling. Moreover, SVA-lncRNA *AK057321* increases expression of each gene increased in human compared to mouse cerebellum: *CHAF1B, IARS, PI4K2B, OSBP,* and *FNIP2* (full-length genes in bacterial artificial chromosomes stably transformed in mouse 3T3 fibroblast cells, Supplementary Fig. 18b). These results provide additional evidence suggesting SVA-lncRNA *AK057321* upregulates genes containing an intronic SVAs to confer species-specific expression patterns in brain.

**SVA-lncRNA *AK057321* opposes the transcription repressing effects of SVA sequences and ZNF91 in a luciferase reporter.** SVA-lncRNA *AK057321* upregulates genes with an intronic SVA in NTera-2 cells where *ZNF91* is also expressed (Fig. 1g–i). *ZNF91* is specific to primates[11,14] and deleting *ZNF91* gene increases expression of SVA-containing transcripts[12]. ZNF91 binds and represses transcription from genes containing SVA sequences[11,12]. We have provided many examples showing SVA-lncRNA *AK057321* and ZNF91 have opposing effects on the expression of genes with intragenic SVAs (see diagram, Fig. 2a): *CDK5RAP2* (Fig. 2b, c), *SCN8A* (Supplementary Fig. 12a–c) and *CHAF1B* (Supplementary Fig. 14b). To examine these effects of the SVA sequences in a simplified system, we examined the effects of SVA-lncRNA *AK057321* and ZNF91 on a luciferase reporter with SVAs from either *CHAF1B* or *HTT*. Both SVAs repress luciferase reporter transcription in human embryonic kidney HEK293T cells (Supplementary Fig. 19a, b). Deleting the VNTR sequence in the SVA eliminates the transcriptional repressive effects of the *CHAF1B* SVA (Supplementary Fig. 19c). SVA-lncRNA *AK057321* partially reverses the repressive effects of the SVAs (*CHAF1B* and *HTT*, Supplementary Fig. 19d, e) and counteracts the further repression produced by ZNF91 (Supplementary Fig. 19e). ZNF91 represses full-length but not VNTR deleted *CHAF1B* SVA (Supplementary Fig. 19f). Intronic SVAs of *JAM2, AGPAT3, POFUT2* and *CDK5RAP2* also repress

transcription of the luciferase reporter compared to an empty vector (Supplementary Fig. 19g). These results indicate SVA sequences repress gene expression, likely by binding to repressor complexes associated with transcription factors like ZNF91. This effect is further magnified by increases of ZNF91. SVA-lncRNA *AK057321* antagonizes the transcriptional repressive effects of SVA sequences that act at least in part by binding transcription factors like ZNF91 that recruits chromatin modifiers to repress gene expression.

**SVA-lncRNA *AK057321* forms RNA:DNA heteroduplexes with genomic SVA sequences in the *CDK5RAP2, SCN8A,* and *CHAF1B* genes it regulates.** We postulated that SVA ribonucleic acid sequences in the SVA-lncRNA *AK057321* might bind complementary SVA deoxyribonucleic acid sequences in the genome. We used two independent approaches (method flow chart in Fig. 7a): (1) an antibody (S9.6) that binds to RNA:DNA heteroduplexes[41] and to a lesser extent AU-rich RNA:RNA duplexes[42]; and (2) an RNA-encoded aptamer tag with affinity to streptavidin added to the sequence of *AK057321*[43] (Supplementary Figs. 20–21).

We examine mouse 3T3 fibroblast clones stably expressing a human full-length *CHAF1B* gene with or without the intronic SVA deleted. SVA-lncRNA *AK057321* upregulates expression of intact but not SVA-deleted *CHAF1B* (Fig. 7b). qPCR primers target *CHAF1B* genomic sequences adjacent to the SVA present in both wild-type and SVA-deleted *CHAF1B* (Fig. 7a). S9.6 RNA:DNA heteroduplex antibody were found to pull down *CHAF1B* genomic sequences when the SVA is present but not when deleted (Fig. 7c). To confirm SVA-lncRNA *AK057321* requires *CHAF1B* SVA sequences for binding, the 5′-aptamer tagged *AK057321* (5′-apt-*AK057321*) was tested in *CHAF1B* clones with or without the SVA. Genomic DNA isolated from each condition was digested with restriction enzymes and precipitated with magnetic streptavidin beads (Fig. 7a). 5′-apt-*AK057321* SVA-lncRNA precipitates *CHAF1B* genomic DNA

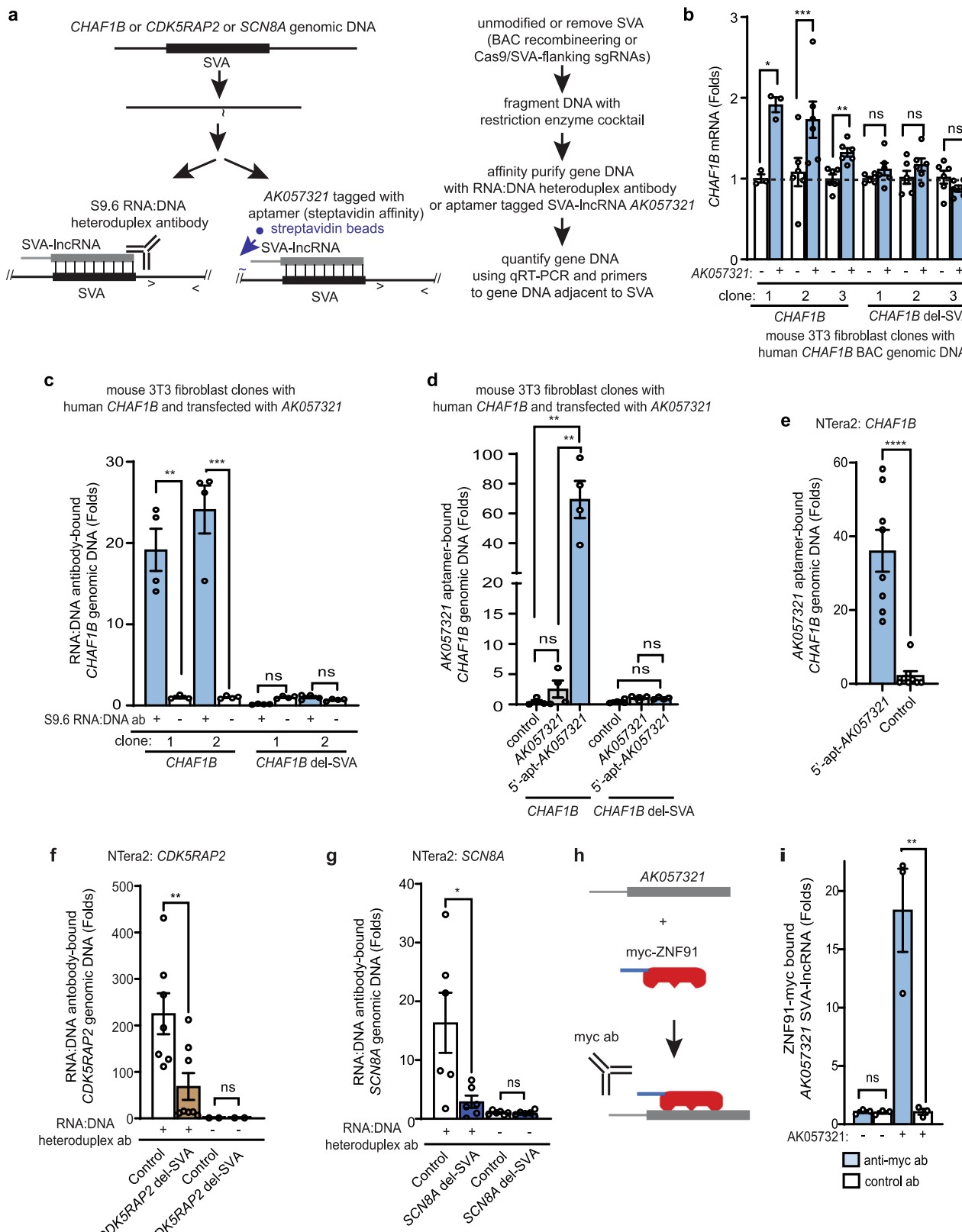

from clones with the SVA present but not those with the SVA deleted (Fig. 7d). In addition, 5′-aptamer tagged *AK057321* co-precipitates the *CHAF1B* genomic DNA in NTera-2 cells (Fig. 7e). Altogether these data show SVA-lncRNA *AK057321* affinity purifies intronic SVA genomic sequences in *CHAF1B* forming a structure recognized by an RNA:DNA heteroduplex antibody.

To provide evidence that SVA-lncRNA *AK057321* binds intronic SVAs in the *CDK5RAP2* and *SCN8A* genes, CRISPR/Cas9-sgRNAs was used to delete the native SVA sequences in NTera-2 cells expressing SVA-lncRNA *AK057321*. qPCR was performed with primers to genomic sequences outside of the SVAs in *CDK5RAP2* and *SCN8A*. S9.6 RNA:DNA heteroduplex

**Fig. 7 SVA-lncRNA *AK057321* forms an RNA:DNA heteroduplex with intronic SVA sequences in *CHAF1B*, *CDK5RAP2*, and *SCN8A* genes and binds transcription factor ZNF91. a** Schematic showing two approaches to isolate genomic DNA associated with SVA-lncRNA *AK057321*. **b** RT-qPCR of human *CHAF1B* mRNA levels in 3T3 clones stably transfected with *CHAF1B* BAC DNA or *CHAF1B* BAC del-SVA DNA and infected SVA-lncRNA *AK057321* lentivirus or control virus. $n = 3$ biological replicates for clone 1, $n = 6$ biological replicates clones 2–3. Unpaired Student's *t* test. **c** qPCR for *CHAF1B* genomic DNA from eluted from RNA:DNA heteroduplex antibody or control antibody pulldowns in *CHAF1B* BAC or *CHAF1B* SVA-del BAC 3T3 clones transfected with *AK057321* cDNA or control. $n = 4$ biological replicates. **d** qPCR for *CHAF1B* genomic DNA in eluted RNA:DNA complexes isolated by streptavidin pulldowns from *CHAF1B* BAC or *CHAF1B* SVA-del BAC 3T3 clones transfected with 5'-apatamer-*AK057321*, untagged *AK057321* or control. $n = 4$ biological replicates. **e** qPCR for *CHAF1B* genomic DNA in eluted RNA:DNA complexes from streptavidin pulldowns performed in NTera-2 cells transfected with 5'-apatamer-*AK057321* or untagged *AK057321* $n = 8$ biological replicates. **c–e** Results are shown as folds enrichment compared to control. **f, g** qPCR for *CDK5RAP2* or *SCN8A* genomic DNA from eluted RNA:DNA heteroduplex antibody and control antibody pulldowns in NTera-2 cells co-transfected with *AK057321* cDNA with or without Crispr/Cas9 sgRNAs to delete the SVA within *CDK5RAP2* (**f**) or *SCN8A* (**g**). $n = 6$ biological replicates each. **h** Approach for co-precipitation of myc-tagged ZNF91 with SVA-lncRNA *AK057321* (Created with Biorender.com). **i** RT-qPCR for *AK057321* RNA levels in eluted anti-myc vs. control antibody pulldowns performed in 293T cells co-transfected with myc-tagged *ZNF91* and either *AK057321* cDNA or control vector. Results are graphed as folds enrichment of *AK057321* compared to control antibody pulldown. $n = 3$ biological replicates. Data represent the mean ± SEM. ****$p < 0.0001$, ***$p < 0.001$, **$p < 0.01$, *$p < 0.05$, ns $p > 0.05$.

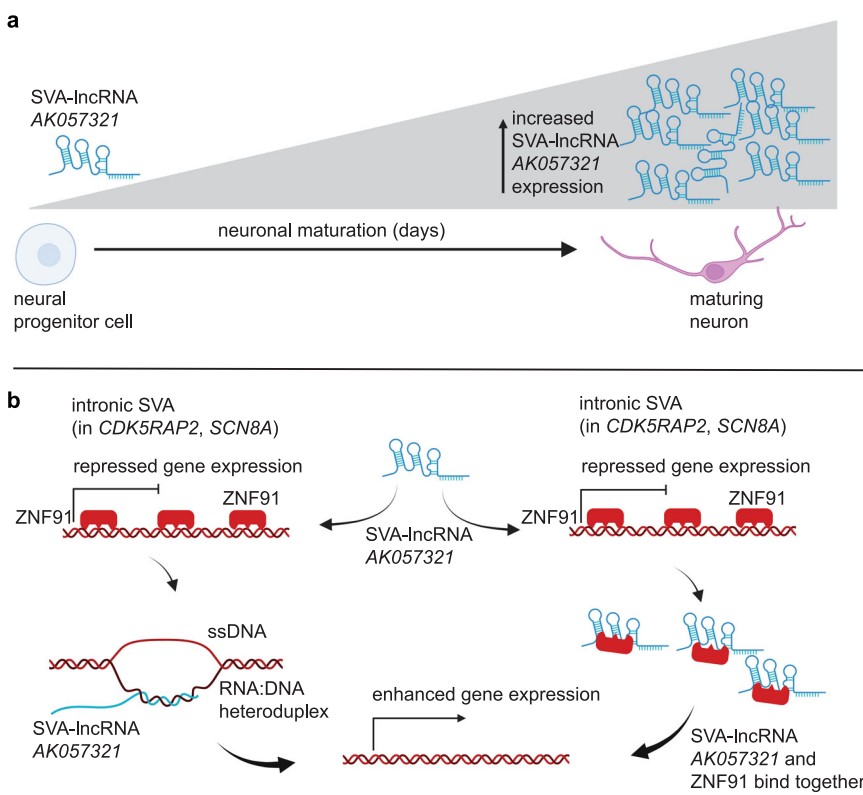

**Fig. 8 SVA-lncRNA *AK057321* expression increases during neuronal maturation and binds intronic SVAs in *CDK5RAP2* and *SCN8A* and ZNF91 transcription factor to release SVA/ZNF91-mediated transcriptional repression and initiate neuronal maturation. a** Schematic showing SVA-lncRNA *AK05732* increases as neural progenitor cells mature into glutamatergic neurons. **b** Schematic where SVA-lncRNA *AK05732* forms an RNA:DNA heteroduplex with intronic SVAs in *CDK5RAP2* and *SCN8A* and binds to the ZNF91 transcription factor that otherwise represses expression of genes containing an intronic SVA (Created with Biorender.com).

antibody co-precipitated *CDK5RAP2* (Fig. 7f) and *SCN8A* (Fig. 7g) genomic sequences. Notably, Cas9-gRNA deletion of the SVA sequences decreases the recovery of *CDK5RAP2* and *SCN8A* genomic sequence (Fig. 7f, g). The results suggest SVA-lncRNA *AK057321* may form RNA:DNA heteroduplexes with intronic SVAs in the *CDK5RAP2* and *SCN8A* genes (see diagram in Fig. 8).

The presence of an entire SVA sequence in SVA-lncRNA *AK057321* (Fig. 1b; exon 3) suggested the possibility that ZNF91 protein, that binds SVA deoxyribonucleic acid sequences in the genome, might also bind SVA ribonucleic acid sequences in SVA-lncRNA *AK057321*. Myc-tagged ZNF91 protein was co-expressed with SVA-lncRNA *AK057321* or control plasmid in

HEK293T cells, immunoprecipitated with anti-myc or control antibodies, and analyzed for SVA-lncRNA *AK057321* by RT-qPCR (methods in Fig. 7h). The results reveal that myc-ZNF91 pulldown enriches for SVA-lncRNA *AK057321* (Fig. 7i). The findings suggest SVA-lncRNA may anneal to the genomic SVA sequences and also bind ZNF91 to disrupt its repressive effects on SVA-bearing genes like *CDK5RAP2* and *SCN8A* thereby promoting neuronal differentiation (biological pathway diagrammed in Fig. 8).

## Discussion
To identify genes repressed by their intronic SVAs to delay neuronal maturation and derepressed by SVA-lncRNA *AK057321*

to drive neuronal maturation, we focused on genes with the following properties: (1) those with human-specific intronic SVAs; (2) that underlie human neurodevelopmental disease; and (3) that are strongly regulated by SVA-lncRNA *AK057321* and ZNF91. This led us to *CDK5RAP2*, a gene with a human-specific tandem SVA-F important to neurogenesis[20–26]. Homozygous inactivating mutations of *CDK5RAP2* underlie human congenital primary microcephaly with reduced cerebral cortex growth[44–46]. We found SVA-lncRNA *AK057321* depletion (shRNA or Cas9/sgRNA) downregulates, while its overexpression upregulates *CDK5RAP2* expression. Bioinformatic genomics studies previously suggested microcephaly genes might underlie the enlarged neocortex of higher primates specifically suggesting a possible role for *CDK5RAP2*[47–49]. Consistent with such ideas, we find the human-specific, tandem SVA_F in the *CDK5RAP2* intron represses its expression to slow neuronal maturation. We further find that SVA-lncRNA *AK057321* induces neuronal maturation by derepressing *CDK5RAP2*.

SVA-lncRNA *AK057321* upregulates a wide variety of neurodevelopmental and neurological disease genes containing intronic SVAs beyond *CDK5RAP2* including *SCN8A*, an epilepsy and intellectual disability sodium channel gene with a human-specific intronic SVA_D. Heterozygous gain-of-function mutations in *SCN8A* underlie early infantile epileptic encephalopathy and heterozygous loss-of-function mutations underlie intellectual disability, suggesting an early dose-dependent role of this gene in neuronal function[28,29]. We find that deleting the intronic SVA_D in *SCN8A* drives selective maturational increases of voltage-gated sodium currents and sodium spikes (immature action potentials) but not other neuronal properties (i.e., hyperpolarized resting membrane potential or increased membrane capacitance). Based on the wide variety of neurodevelopmental disease genes containing intronic SVAs that are regulated by SVA-lncRNA *AK057321*, we suggest intragenic SVAs may delay maturation of multiple specialized neuronal functions (e.g., synaptogenesis involving *NRXN1*, *NRXN2*, and *SHANK2* and the developmental change of chloride reversal potential that converts GABAergic transmission from excitatory to inhibitory involving *WNK3*).

We show the neuronal maturation-promoting effects of SVA-lncRNA *AK057321* are mimicked by decreasing expression of the SVA-repressive transcription factor ZNF91. These data suggest a biological pathway where SVA-lncRNA *AK057321* opposes the transcriptional repressive effects of ZNF91 repressor complex bound to the intronic SVA sequences. KZNFs such as ZNF91 transcriptionally repress genomic regions containing retrotransposons by binding TRIM28[50] leading to repressive epigenetic modifications[51,52] with some components mediated by the human silencing hub (HUSH) complex[16].

We provide evidence that SVA ribonucleic acid sequences in SVA-lncRNA *AK057321* may form an RNA:DNA heteroduplex with complementary genomic SVA deoxyribonucleic acid sequences in the introns of *CDK5RAP2*, *SCN8A* and *CHAF1B* genes to increased their expression. SVA-lncRNA *AK057321* also binds ZNF91 suggesting a possible decoy effect that might displace ZNF91 transcriptional repressor complex from genomic SVA sequences. Taken together, these data reveal the existence of an SVA transposon-based gene regulatory system involving intronic SVAs bound by either ZNF91 repressor complex to repress, or by SVA-lncRNA *AK057321* to enhance gene expression, thereby regulating the timing of neuronal maturation in the hominoid species lineage.

Human Huntingtin disease gene *HTT*[39] contains an intronic SVA. We find *HTT* gene expression is enriched in cortex and cerebellum relative to brainstem in human (not seen for native mouse *Htt* gene or human *HTT* transgene in mouse brain) and that this cortex and cerebellum-enriched expression pattern is driven by SVA-lncRNA *AK057321*. Using mice transgenic for human SVA-lncRNA *AK057321* and human chromosome 21, we show that SVA-lncRNA *AK057321* upregulates expression of genes that harbor intronic SVAs on human chromosome Chr21 in mice[40], increasing *POFUT2*, *CHAF1B* and *KCNJ6* genes but not mouse orthologs lacking SVAs. These results provide evidence that SVA-lncRNA *AK057321* may increase the expression of multiple neurodevelopmental genes with intronic SVAs in the cortex and cerebellum relative to subcortical brain regions. We speculate that increasing expression of neurodevelopmental genes in cortex and cerebellum glutamatergic neurons relative to the other brain regions might enable these neurons to become the drivers of subcortical neuronal circuit activities (e.g., in thalamus, hypothalamus, and brainstem).

Polymorphic SVA insertions found in the human population in protein-coding or regulatory regions have resulted in insertional mutagenesis or aberrant RNA splicing to cause disease. Examples include intragenic SVA insertions in Fukuyama muscular dystrophy[53], X-linked dystonia Parkinsonism[54], Neurofibromatosis type 1[55], and autosomal recessive hypercholesterolemia[56]. *De novo* SVA insertions have also been reported in non-coding regions of the genome including introns[57]. Here we provide insights into the functional significance of such intronic SVAs acting to repress gene expression during neuronal maturation and in subcortical brain regions in adulthood. We speculate that newly acquired intronic SVAs might promote previously unrecognized genetic forms of neurodevelopmental, neurological, or psychiatric disease.

We find SVA-lncRNA *AK057321* is expressed in glutamatergic but not cholinergic or dopaminergic neurons. We speculate that other SVA-lncRNA family members might be restricted to other neuronal and glial cell types enabling independent regulation of the timing of gene expression in those cells.

SVA-lncRNA *AK574321* is also expressed in testis and multiple genes regulating spermatogenesis and fertility contain intronic SVAs (e.g., *SPATA5*[58,59] and *CDK5RAP2*[60]). This observation suggests the SVA transposon-based gene regulatory system might influence species-specific aspects of male fertility and new intronic SVAs acquired during speciation may confer fertility advantages to those species.

Given the numerous genes with intronic SVAs in hominoids (many specific to human), future studies should examine the functional role of the other intronic SVAs in regulating glutamatergic neuron progenitor growth, maturation, structure and function. Such studies could provide further insights into how this SVA transposon-based gene regulatory system may have helped shape structures and functions unique to the human brain.

## Methods

**Statistics and reproducibility**. Statistical analysis of datasets showing unpaired two-tailed Student's *t* test, one-way ANOVA with Bonferroni post hoc tests, and two-way ANOVA with Bonferroni post hoc tests were calculated on datasets in this study using Graphpad Prism 9.3.1 software. Graphed data represent the mean ± SEM.$****p < 0.0001$, $***p < 0.001$, $**p < 0.01$, $*p < 0.05$, ns $p > 0.05$ versus control unless otherwise indicated. Blinding was performed during data collection and analysis. All cell, animal, and human samples were randomized. Group sizes were chosen based on adequate statistical power to detect differences. Details of biological replicates described in figure legends. No data was excluded. Experiments were independently replicated at least twice (with the exception of Fig. 1g (a subset of genes), Fig. 7i and Supplementary Fig. 3). For mice, a biological n indicates tissue from an independent animal with the same genotype. For cells, a biological n indicates cells from an independent set of transfection/FACS sort with the same vectors.

**GO term enrichment analysis**. GO term enrichment analysis was performed at the following websites: pantherdb.org and g:Profiler – a web server for functional enrichment analysis and conversions of gene lists (ut.ee).

**NTera-2 transfection and cell sorting**. NTera-2, CRL-1973, was purchased from ATCC. NTera-2 cells were plated at a density of 5000 cells/cm$^3$ 12–16 h prior to transfection with the indicated plasmids using Lipofectamine 3000 (ThermoFisher) for the indicated times. Cells were harvested and FACS sorting of NTera-2 transfected fluorescent cells was performed at the BIDMC Flow cytometry Core on a MOFLO Astrios EQ and on a BD FACSAria II. Each experiment utilized a fluorescent empty vector control, and un-transfected NTera2 cells were used as the negative control to set the gate for positive cell collection in each experiment.

**NTera-2 neuronal maturation with mouse astrocyte co-plating**. For *Ngn2*-induced differentiation, two lentiviruses were each produced from expression plasmids pLV-TetO-hNGN2-eGFP-Puro (Addgene 79823) and FUGW-UbCp-rtTA3 (Addgene 105171) using the method of co transfection HEK293T cells with psPAX2 (Addgene 12260) and pMD2.g (Addgene 12259) (Addgene: Lentivirus Production Protocol). For all co-culture studies with transfected NTera-2 cells, mouse glial cells were isolated from the forebrains of 1-3 day-old CD1 pups as described[61]. Isolated forebrains were put into calcium and magnesium-free HANKS (ThermoFisher 14175015) and dissociated with trypsin-EDTA (ThermoFisher 25200056). After 20 min, trypsin was inhibited by suspension of the cells into MEM (ThermoFisher 51200038) supplemented with 10% Fetal Bovine Serum (ThermoFisher). Cells were triturated through a pipette and passed through a 0.40 μM cell strainer, pelleted and washed twice, re-suspended and then cultured in T75 flasks. Media was changed the next day and cells were passaged one time to avoid neuronal contamination before their use in co-culture experiments. For co-culture experiments of *Ngn2*-induced NTera-2 cells with mouse astrocytes, NTera-2 cells were transduced similarly as described[62] with the following daily steps. On day −5, glass coverslips were coated with Matrigel (BD Biosciences 354230) within wells of a 24-well plate and on day −4, isolated glial cells were trypsinized and plated onto coverslips with matrigel and set aside until they reached 70–80% confluency on day 2. On day −2, Ntera-2 cells were plated at 30-50% confluency in T75 flasks. On day 1—NTera-2 cells were co-infected with *Ngn2* and rtTA containing lentiviral supernatants as prepared above, in DMEM supplemented with polybrene (8 μg/mL, MilliporeSigma). On day 0, doxycycline was added at a concentration of 2 μg/mL to induce *Ngn2* and puromycin resistance gene expression. On Day 1 puromycin (ThermoFisher A1113803) was added at a concentration of 1 μg/mL to select cells expressing Ngn2. On day 2, Ngn2-positive NTera-2 cells were removed from T75 flasks with TrypLE (ThermoFisher 12604013) and were co-plated at a density of 1.5 × 10$^4$ cells per well in Neurobasal medium (ThermoFisher 21103049) supplemented with Glutamax (ThermoFisher 35050079), B27 (ThermoFisher 17504044) and BDNF (R&D Systems 248-BD) along with 2 μg/mL doxycycline. Ara-C (2 g/l, Sigma) was added to inhibit astrocyte proliferation once astrocytes reached confluency. 50% of the media (0.3 ml) was changed every 3–4 days until experiments were performed. In experiments lacking *Ngn2*, FACS-sorted NTera-2 cells were plated at a low density (5000 cells per cm$^2$) along with mouse astrocytes onto matrigel-coated glass coverslips in Neurobasal medium supplemented with B27, Glutamax and BDNF. Ara-C (2 g/l, MilliporeSigma) was added to inhibit astrocyte proliferation once astrocytes reach confluency. 50% of the media (0.3 ml) per well was changed every 2-3 days.

**Immunostaining**. *Ngn-2* induced neurons at 90 days after co-plating were fixed in 4% PFA in PBS for 1 h at room temperature. After PBS washing, anti-beta 3 Tubulin (TuJ1) Antibody (2G10, Saint Cruz, 1 to 2000 dilution) was used for staining TuJ1 in blocking solution at 4 °C temperature overnight. Sections were washed and incubated with Alexa-conjugated secondary antibodies (1:500, Invitrogen) for 3 h at room temperature. Cells then were mounted in Vectashield with DAPI (Vector). Fluorescent images were taken using a LSM510 confocal microscope (Zeiss) and were processed with Image J. For immunostaining of neuronal maturation due to *AK057321* overexpression, NTera-2 cells were FACS-sorted NTera-2 cells, *AK057321* overexpression vs. control cells, were individually co-plated with mouse astrocytes for 13 days as described above. Cells were then fixed in 4% PFA in PBS, followed by permeabilization with 0.3% Triton X-100, blocking in 5% goat serum/PBS for 1 h. MAP2 antibody (1:100, MilliporeSigma AB5622) was diluted in 2.5% goat serum in PBS and incubated with cells for 1 h at room temperature, washed three times in PBS, followed by incubation with secondary antibody (1:500, ThermoFisher A-11011) for 1 h at room temperature. After washing in PBS, the cells were mounted with anti-fade medium (Vectorshield plus H-1900-10) and stored at 4 degrees overnight prior to microscopy with LSM510 confocal microscope (Zeiss). Images were processed with Image J.

**Sholl analysis**. SVA-lncRNA *AK057321* (or control) transfected NTera-2 cells, matured for 13 days with mouse astrocytes on matrigel-coated glass coverslips, were fixed in 4% PFA in PBS for 10 min at room temperature, followed by 3 times wash with PBS, and mounted with anti-fade medium (Vectorshield plus; Cat. #: H-1900-10) and placed at 4 degrees overnight before subjected to immuno-fluorescence microscopy. Images were acquired with Leica DM6 confocal microscopy by Leica Application Suite X (LAS X) software. Exposure time was fixed throughout the imaging process. Images were converted to 8-bit format and processes were traced using NeuroJ plugin. The center of the cell body was used as a

reference point and 94 concentric circles were generated on the tracings: the starting radium was 30 pixels (9.2 μm) and the ending radium was 380 pixels (121.6 μm) at the step of 5 pixels (1.6 μm). The number of intersections at each concentric circle was registered and plotted. The total length of processes, mean length of processes, the average maximum/ minimum length of processes, number of processes were quantified from the tracing files. Cell morphology was quantified with ImageJ plugins for circularity. Cell area was measure with ImageJ with fixed threshold.

**Electrophysiology**. Recordings were made on the indicated days after NTera-2 and mouse astrocytes were co-plated on matrigel and poly-lysine-coated glass coverslips immediately after sorting. Membrane potential and currents were recorded using the whole-cell configuration of the patch-clamp technique with an EPC-10 HEKA amplifier and the HEKA PatchMaster 9 data acquisition software. The extracellular solution consisted of (in mM): 140 NaCl, 5.4 KCl, 0.5 MgCl$_2$, 2.5 CaCl$_2$, 5.5 HEPES, 11 glucose, and 10 sucrose, pH 7.4 with NaOH for current clamp mode. Resistance of the patch pipette was 3–4 MΩ when filled with (in mM) 105 K-gluconate, 20 KCl, 1 CaCl$_2$, 5 MgATP, 10 HEPES, 10 EGTA, and 25 glucose, pH 7.4 with KOH. Action potential was elicited by a 100 ms square current injection from 0- 600 pA with 100 pA increment. Cells were held at −90 mV by hyperpolarizing currents to induce sodium-based action potential. ΔV was calculated by measuring the difference between peak amplitude and 100 ms after current injection. To record Na$^+$ currents, the extracellular solution was changed to a solution consisting of (in mM): 130 NaCl, 20 TEA-Cl, 1 CaCl$_2$, 5 MgCl$_2$, 10 HEPES, and 5.56 glucose, pH 7.4 with NaOH. Internal solution was changed to (in mM) 10 NaCl, 130 CsCl, 5 MgCl$_2$, 5 EGTA and 10 HEPES. Na$^+$ currents were evoked from a holding potential of −90 mV by stepping to voltages −30 mV for 20 ms. Series resistance of 5–10 MΩ was electronically compensated 80-90%. Current traces were sampled at 10 kHz and filtered at 5 kHz. Comparisons were made at day 15 unless mentioned otherwise.

**SVA-lncRNA *AK057321* expression and knockdown**. SVA-lncRNA *AK057321*, Homo sapiens cDNA FLJ32759 fis, clone TESTI2001793, was subcloned into BamHI and EcoRI sites of pLVX-mcherry, and also into XbaI and BamHI sites of BioSettia pLV-09. For knockdown of SVA-lncRNA *AK057321*, target sequences listed in Supplementary Table 1 were cloned into a modified version of pLVX-shRNA2-mcherry in which the CMV promoter was excised with BamHI and AgeI digestion and replaced with the UbC promoter, P2A, the zeomycin resistance gene and T2A in frame with mcherry using HiFi DNA Assembly (primers listed in Supplementary Table 1). UbC, P2A, and T2A sequences were PCR-amplified from pLV-TetO-hNGN2-eGFP-Puro (Addgene #79823) and the zeocin resistance gene sequence was amplified by PCR from FRT-Em7-Zeo-FRT plasmid (Thermo Fisher).

**myc-tagged human *ZNF91***. ORF clone expressing myc-tagged human *ZNF91*, accession number NM_003430.3 was purchased from GeneCopeia.

**CRISPR/Cas9-mediated deletions**. For CRISPR-Cas9 deletion of *ZNF91*, published gRNAs[12] were cloned into a modified version of lentiCRISPRv2 hygro (Addgene #98291) in which the hygromycin resistance gene was excised by digestion with PshAI and MluI and replaced with PCR-amplified mcherry from pLVX-shRNA2-mcherry using NEB HiFi DNA assembly (primers listed in Supplementary Table 2). All PCR amplifications for cloning were performed with Phusion (NEB; with cycling conditions: 1, 98 °C for 30 s, 25–35, 45–72 °C for 10–30 s, 72 °C for 15–30 s per Kb, last cycle, 72 °C for 5–10 min). *ZNF91* sgRNAs were first cloned individually into the BsmBI site of lentiCRISPRv2 hygro. To ensure delivery of both sgRNAs into every transfected cell, both sgRNAs were cloned in tandem into lentiCRISPRv2-mcherry, each with their own U6 promoter, using long primers to amplify one sgRNA and short primers for the other. lentiCRISPRv2-mcherry with no sgRNAs was digested with KpnI and NheI and each guide was PCR-amplified from the corresponding Addgene #98291 vector (lentiCRISPRv2 hygro) it was originally cloned into. The digested lentiCRISPRv2-mcherry vector (KpnI and NheI), along with PCR-amplified U6-sgRNA (up; long_guide) and PCR-amplified U6-sgRNA (down; short_guide) were assembled using NEB HiFi DNA assembly (primers listed in Supplementary Table 2). For CRISPR-Cas9 deletion of the tandem SVA located within the *AK057321* gene, candidate sgRNAs (located 5′ and 3′ to the SVA sequence) were each individually cloned into lentiCRISPRv2 hygro using the recommended protocol: lentiCRISPRv2 puro was a gift from Brett Stringer (Addgene plasmid # 98290; http://n2t.net/addgene:98290; RRID:Addgene_98290). Cloned sgRNAs were tested in pairs (5′ sgRNA plus 3′ sgRNA) to determine an optimal combination to achieve SVA deletion. These sgRNAs (5′ sgRNA plus 3′ sgRNA) were then cloned in tandem using the same HiFi DNA assembly strategy as was done for lentiCRISPRv2-mcherry, but into a modified version of lentiCRISPRv2 hygro in which T2A and blue fluorescent protein (BFP) were inserted in frame after the MluI site in lentiCRISPRv2 hygro, (primers listed in Supplementary Table 2). sgRNAs used to delete each SVA located within introns of *CDK5RAP2*, *SCN8A* and *CHAF1B* were similarly cloned and tested, with optimized sgRNA pairs cloned in tandem into lentiCRISPRv2-BFP cut with KpnI and NheI. Supplementary Table 3 lists the

sgRNA sequences used in this study (PAM sequences are underlined). "Left" vs. "right" refers to sgRNA sequences used to cut on each side of the SVA; one sgRNA on the left (5′ to the SVA) and one sgRNA on the right (3′ from the SVA). Supplementary Table 4 lists the genomic PCR primers used to detect the CRISPR-Cas9 deletion of SVAs located within the *AK057321* lncRNA sequence, and within *CDK5RAP2*, *SCN8A* and *CHAF1B* introns. Genomic PCRs were performed using Azura 2X Taq Red Mix, using cycling conditions: 1; 95 °C, 1 min, 25–40; 95 °C, 15 s, 60 °C, 15 s, 72 °C, 30 s per Kb).

**RNA aptamer tag added to SVA-lncRNA *AK057321***. A sephadex and a streptavidin dual aptamer tag was constructed by PCR with overlapping oligos and added to SVA-lncRNA-AK057321 at the 5′ end with the spacer sequence "GGAAGAGGAAGA" after the tag and before the start of the *AK057321* sequence, internally between exons 2 and 3 with the same spacer sequence flanking the 5′ and 3′ end of the dual tag sequence, and at the 3′ end with the spacer sequence after the *AK057321* sequence and before the start of the dual aptamer tag sequence in Supplementary Table 5.

**CDK5RAP2 knockdown and overexpression**. *CDK5RAP2*-set of piLenti (Human) siRNAs: piLenti-siRNA-GFP that targets four *CDK5RAP2* mRNA sequences were purchased from abm, catalog number i004292. Inducible Expression of *CDK5RAP2*: Overexpression of *CDK5RAP2* has reported to be toxic, so an inducible construct to express lower levels of *CDK5RAP2* was generated by us. For inducible expression of *CDK5RAP2*, the CMV promoter within pRcCMV Cep215 (Addgene #41152) was excised with PciI and AfeI digestion. The TetO sequence was amplified using TetO FWD and TetO REV primers listed in Supplementary Table 6 from pLV-TetO-hNGN2-eGFP-Puro (Addgene #79823) vector and was inserted into digested pRcCMV Cep215 using NEB HiFi Assembly with HiFi PCR primers listed in Supplementary Table 6. For expression studies, TetO-*CDK5RAP2* was co-transfected with FUW-rtTA (Addgene #20342) Ntera-2 cells and *CDK5RAP2* expression was induced with 1 μg/mL doxycycline for 24 h.

**Luciferase assays**. All luciferase assays were performed in HEK293T cells. After 48 h cells were assayed for luciferase activity using ONE-Glo EX Luciferase Assay System (Promega) on a BioTek Synergy luminometer using Gen5 software. The *CHAF1B* SVA was amplified from bacterial artificial chromosome (BAC) clone RP11-108J14 purchased from BACPAC https://bacpacresources.org/order_clones.php using Phusion polymerase (NEB) the *CHAF1B* FWD and *CHAF1B* REV primers listed in Supplementary Table 7. All PCR amplifications for SVA cloning were performed with Phusion (NEB; with the GC buffer and DMSO). Cycling conditions: 1, 98 °C for 30 s, 25–35, 45–72 °C for 10–30 s, 72 °C for 15–30 s per Kb, last cycle, 72 °C for 5–10 min). The SVA was then cloned into the KpnI and NheI sites within the pGL3-luciferase promoter vector (Promega) using PCR fragment amplified with the FWD and REV HiFi primers (Phusion) with HiFi NEBuilder Assembly (NEB). Positive clones were selected and screened for the presence of the SVA by PCR with Phusion (NEB) using the *CHAF1B* SVA FWD and REV primer sets. The *CHAF1B*minusVNTR version of the *CHAF1B* SVA was assembled by PCR from two separate pieces (Part A and Part B) that lacked the VNTR region, each PCR-amplified from bacterial artificial chromosome (BAC) clone RP11-108J14 using Phusion polymerase (NEB). The Part A and Part B SVA pieces were then cloned into the KpnI and NheI sites within the pGL3-luciferase promoter vector (Promega) using PCR fragments amplified with the FWD and REV HiFi primers (Supplementary Table 8) with HiFi NEBuilder Assembly (NEB). Positive clones were selected and screened for the presence of the SVA by PCR with Phusion (NEB) using the *CHAF1B* SVA FWD and REV primer sets (Supplementary Table 7). The *HTT* SVA was amplified by gradient PCR using Phusion (NEB) from the bacterial artificial chromosome (BAC) clone RP11-866L6 purchased from BACPAC https://bacpacresources.org/order_clones.php., using the *HTT* SVA FWD and REV primers listed in Supplementary Table 9. The PCR fragment containing the *HTT* SVA was then TA-cloned into TOPO pcR2.1 (Invitrogen) and positive clones were selected with screening primers listed in Supplementary Table 9. The *HTT* SVA-containing fragment was then excised from TOPO pcR2.1 with KpnI and XhoI and cloned directly into these sites of KpnI/XhoI digested pGL3-luciferase promoter vector (Promega). Positive clones were screened for the screening primers listed in Supplementary Table 9. *JAM2*, *AGPAT3*, *POFUT2*, and *CDK5RAP2* SVAs clonings: SVAs were amplified from individual bacterial artificial chromosome (BAC) clones obtained from BACPAC https://bacpacresources.org/order_clones.php using Phusion polymerase (NEB) the primers listed in Supplementary Table 10. The SVAs were then similarly cloned into the KpnI and NheI sites within the pGL3-luciferase promoter vector (Promega) using PCR fragment amplified with the FWD and REV HiFi primers (Phusion) with HiFi NEBuilder Assembly (NEB) using primers listed in Supplementary Table 10. Positive clones were selected and screened for the presence of the SVA by PCR with Phusion (NEB) using the CHAF1B SVA FWD and REV primer sets.

**BAC recombineering**. BAC recombineering was used to insert neomycin resistance gene into various BAC clones and generation of stable NIH3T3 clones. Genomic DNA bacterial artificial chromosome (BAC) clones were purchased from

was purchased from BACPAC (RP11) and Invitrogen (CTD), https://bacpacresources.org/order_clones.php and https://clones.thermofisher.com as listed in Supplementary Table 11. These BACs are on the pBACe3.6 backbone (RP11) or on the pBeloBAC11 backbone (CTD). A a neomycin resistance gene cassette was added to each BAC in Supplementary Table 12. The neomycin resistance gene was amplified by PCR with Phusion (NEB) using NeomycinR FWD and REV primers from the PGK-Em7-Neo plasmid. All primer sequences used in recombineering are shown in Supplementary Table 12. For the RP11 and CTD clones, PCR-amplified arms were generated and selection primers as shown in Supplementary Table 12. The neomycin cassette was combined with these arms by PCR creating an amplified fragment containing the upstream arm – neomycin resistance gene – and downstream arm. This PCR fragment was gel-purified and inserted into the clones by homologous recombination using electrocompetent SW105 cells. Cells were then plated on chloramphenicol/kamamycin plates for selection. Individual clones were screened for insertion with the following primers that amplify on the BAC backbone around the loxP site listed in Supplementary Table 13. BAC recombineering to delete the SVA in a CHAF1B BAC clone: The BAC clone RP11-108J14, chr21:36384336-36558420 (174,085 bp; Hg18 UCSC Genome Browser) contains the entire human *CHAF1B* genomic sequence, including an intragenic SVA was first modified by the addition of a neomycin drug resistance cassette. To replace the SVA sequence within the *CHAF1B* on this BAC clone, PCR (Phusion polymerase; NEB) was used to generate three pieces (Supplementary Table 14): a left arm with sequence identity to the BAC upstream of the SVA, a right arm with identity to the BAC downstream of the SVA and then a middle piece that contains the zeocin drug resistance cassette and Frt sites, from FRT-Em7-Zeo-FRT plasmid. BAC recombinations were performed using electrocompetent SW105 with 200-300 of gel-purified assembled DNA, followed by plating on chlro/kan/zeo plates. Clones were expanded and tested for insertion by PCR from the bacteria cultures using a zeo oligo and a *CHAF1B* oligo (Supplementary Tables 8 and 15). The zeocin resistance gene was amplified by PCR from FRT-Em7-Zeo-FRT clone GF-21 with the primers listed in Supplementary Table 15. In summary, the overall strategy generated three pieces, a left arm with sequence identity to the BAC upstream of the SVA, a right arm with identity to the BAC downstream of the SVA and then a middle piece that contains the zeo drug resistance cassette using the primers listed in Supplementary Tables 14–15. These three overlapping pieces were then assembled in the PCR machine using Phusion polymerase (NEB).

**Gene expression analysis**. Total RNA was isolated using Trizol (Fisher Scientific). All RNA pellets were resuspended and stored in RNASecure (Fisher Scientific) at −80 °C. Reverse transcription of RNA was performed after DNAse treatment (NEB) with M-MuLV Reverse Transcriptase (NEB). Quantification of mRNA levels was performed using PowerUp SYBR (Applied Biosystems; 50 °C 2 min, 95 °C 2 min, 45 cycles of 95 °C for 3 s and then 60 °C for 20 s) or Taqman Gene Expression master mix (Applied Biosystems; 50 °C 2 min, 95 °C 10 min, 45 cycles of 95 °C for 15 s and then 60 °C for 1 min) on a BIO-RAD CFX384 Real-Time System. Primers and probes were ordered from Integrated DNA technologies (IDT, Supplementary Table 16). Other primers used for expression analysis were ordered from Eurofins (Supplementary Tables 17 and 18). Primers that recognize both human and mouse transcripts (data shown in Supplementary Fig. 18) are bolded in Supplementary Table 17. Supplementary Table 18 lists the primers used for relative-quantitative PCR (rqPCR) of *AK057321* expression (Supplementary Fig. 16). Supplementary Table 19 lists the thermocycler conditions for reverse transcriptase and polymerase chain reactions. Multiple bands were amplified in the PCR from transgenic mice and human cDNA samples with different primer sets. The band at the expected size of *AK057321* amplicons and two additional bands were isolated from agarose gel with QIAquick Gel Extraction Kit (Qiagen, 28704) and sequenced at the Beth Israel Deaconess Medical Center DNA Sequencing Core. The band that was identified as *AK057321* was used in relative-quantitative PCR experiments to determine expression levels in transgenic mouse and human brain regions. Classic and Universal QuantumRNA 18 S Internal Standard kits (Ambion, AM1716 and AM1718) were used according to manufacturer's instructions. PCRs were performed with Phire Hot Start DNA polymerase (Finnzymes F-120) on cDNA prepared from brain tissue of *AK057321* transgenic mice (1 or 2 copies of the transgene, 3 mice for each genotype) or human brain tissue. Optimization reactions were performed to determine the linear range and cycle number of PCR amplification, and 18 S Primer:Competitor ratio. The reactions were run on a DNA agarose gel, and images were acquired with a BioRad ChemiDoc XRS + system. The band intensities were quantified with ImageJ software. RNA from iPSCs and each differentiated neuron type was obtained from Elixirgen Scientific, https://www.elixirgensci.com.

**Transgenic mice**. For generation of AK057321 transgenic mice, genomic DNA bacterial artificial chromosome (BAC) clone RP11-1043N4 32,604,995-32,769,907 (164,912 bp; Hg18 UCSC Genome Browser) was purchased from BACPAC (https://bacpacresources.org/order_clones.php). BAC DNA was prepared by double sodium acetate precipitation and CsCl₂ gradient method and confirmed by sequencing analysis. BAC DNA was linearized by restriction enzyme PI-SceI (NEB) and then microinjected in the BIDMC transgenic core facility. The studies use human mutant HTT transgenic mice bearing an HD model mutation and carried in a yeast artificial chromosome vector, YAC128 mice (Jackson Laboratory, mouse

strain #004938). For generation of CHAF1B transgenic mice, genomic DNA bacterial artificial chromosome (BAC) clone RP11-108J14, chr21:36384336-36558420 (174,085 bp; Hg18 UCSC Genome Browser) contains the entire human CHAF1B genomic sequence, including an intronic SVA (BACPAC, https://bacpacresources.org/order_clones.php). BAC DNA was prepared by purification with Nucleobond BAC 100 kit (Machery-Nagel). BAC DNA was linearized by restriction enzyme PI-SceI (NEB) and then microinjected in the BIDMC transgenic core facility. All protocols were approved by the BIDMC IACUC and Harvard Medical Area Standing Committee on Animals.

**Sample tissue preparation.** Adult mice (between 2-6 months in age) were deeply anesthetized with isoflurane and rapidly decapitated. For studies involving human mutant HTT transgenic and matched controls, males were chosen. For studies involving human Chr21 transchromosomic mice and human CHAF1B transgenic mice, animals were chosen without regard to sex. The brain was removed and cortex, cerebellum, hippocampus, subcortical, and brain stem were microdissected with fine scissors and forceps. Samples were flash frozen on dry ice and stored at −80 °C until RNA was prepared. Human brain tissue was collected randomly from sequential postmortem cases with next-of-kin consent in the Beth Israel Deaconess Medical Center Pathology Department with IRB approval. Samples from males (50-65 years of age) cortex (frontal, temporal, parietal, occipital), cerebellum (cortex and vermis), hippocampus, caudate, thalamus and pons were collected, flash frozen on dry ice, and stored at −80 °C until RNA was prepared.

**SVA-lncRNA AK057321 binding to genomic SVA studies.** For SVA-lncRNA AK057321 binding to the intronic SVA in CHAF1B, NIH3T3 clones stably expressing with the human CHAF1B gene from genomic BAC RP11-108J14, or from a version of genomic BAC RP11-108J14 in which the SVA was deleted were first used. For SVA-lncRNA AK057321 binding to the intronic SVA in endogenous CHAF1B, CDK5RAP2 and SCN8A genes, NTera-2 cells were used. After each transfection, cell pellets were harvest and flash frozen at −80 °C. Genomic DNA was extracted from each cell pellet using phenol/chloroform/isoamyl alcohol (FisherSci) and genomic DNA was washed with 70% ethanol, dried and resuspended in RNASecure (InVitrogen). 20 µg of genomic DNA from each condition were digested with an enzyme cocktail containing BamHI, HindIII, XbaI, PstI and purified RNAse inhibitor. The RNAse inhibitor (Addgene #153314) was grown in bacteria and was purified using NEB's pMAL Protein Fusion and Purification System, E8200. To assess RNA:DNA heteroduplex formation, each sample was resuspended in TE buffer with 0.1% NP-40 equivalent detergent (FisherSci) and immune-precipitated with S9.6 or control antibody with Protein G magnetic beads (FisherSci) overnight, supplemented with RNAse inhibitor. Beads were then washed 5 times in buffer supplemented RNAse inhibitor. Complexes were eluted by heating the washed beads at 80 °C for 5 min. The presence of the genomic DNA adjacent to the respective intronic SVAs (CHAF1B, CDK5RAP2, SCN8A) are assessed by qPCR using the genomic primer and probes listed in Supplementary Table 20. For pulldown of 5′-aptamer tagged SVA-lncRNA AK057321, the same protocol was followed except pulldowns were performed with magnetic strepta-vidin beads (FisherSci) for only 1 h.

**SVA-lncRNA AK057321 binding to ZNF91 protein.** myc-tagged ZNF91 and SVA-lncRNA AK057321 were co-expressed in 293T cells for 48 h, harvested and then cell pellets were flash frozen at −80 °C. Cells were lysed and immune-precipitated with anti-myc or control antibodies for 2 h in the presence of RNAse inhibitor, followed by 5 washes with buffer and then elution of complexes by heating at −80C for 5 min. RT-qPCR was performed to detect the presence of SVA-lncRNA AK057321 in each condition using the probes listed in Supplementary Table 16.

**Reporting summary.** Further information on research design is available in the Nature Portfolio Reporting Summary linked to this article.

## Data availability
All data generated or analyzed during this study are included in this published article, and Supplementary Data 2–3 contain source data underlying supplementary and main figures, respectively.

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

## Acknowledgements

The authors thank Anderson Lab members Oriana DiStefano, Greg Salimando, Rebecca Broadhurst and Scott Rochard for mouse colony upkeep and genotyping. We thank David Stoppel for mouse brain dissections and Annmarie McKeon for SVA genome annotation. We thank the BIDMC Flow Cytometry Facility, the Harvard Center for Biological Imaging at Harvard University, the Neurobiology Imaging Facility at Neurobiology Department of Harvard Medical School for consultation and instrument availability that supported this work (in part through a NINDS P30 Core Center grant #NS07203) and Boston Children's Hospital IDDRC (1U54HD090255, P30HD18655). This work was supported by funding to M.P.A. from The National Institute of Mental Health (R01MH112714, R01MH114858, and 1R21MH100868), The National Institute of Neurological Disorders and Stroke (1R01NS08916), The Eunice Kennedy Shriver National Institute of Child Health and Human Development (1R21HD079249), The Nancy Lurie Marks Family Foundation, Landreth Foundation, Autism Speaks/National Alliance for Autism Research, and the Simons Foundation.

## Author contributions

M.N., W.C., E.O., Y.H., Y.N., M.J., M.B., A.M., and M.P.A. designed the study. M.N and M.P.A wrote the manuscript. M.N., W.C., E.O., Y.N., Y.H., M.J., M.B., A.M. performed all the experiments and analyses except for the following: W.C. and Y.N. performed the electrophysiology.

## Competing interests

Y.H., Y.N., and M.P.A. became employees of Regeneron Pharmaceutical. The other authors declare no competing interests.
