## [Peer Review File · Communications Biology]

Reviewers' comments:

Reviewer #1 (Remarks to the Author):

In humans, SINE-VNTR-Alu (SVA) elements are one of three still active transposable element (TE) families. Using pluripotent human teratoma-derived cell line NTera-2 and three transgenic mice (HTT, AK057321 and CHAF1B), Nadler et al. found that SVA-lncRNA AK057321 can bind SVAs in the introns of several neuronal genes, such as microcephaly gene CDK5RAP2 and infantile epileptic encephalopathy gene SCN8A. SVAs suppress the expression of these genes, while AK057321 promote their expression. AK057321 knockdown by shRNA or Cas9-sgRNA, or over-expression experiments also shown that it suppresses pluripotency genes (e.g., NANOG and POU5F1).

The author suggested this AK057321/SVAs network is important for neurodevelopment and neoteny of the hominid brain. Overall, this is a comprehensive and interesting study, that employed multiple advanced technologies including transgenic mice, iPSC, gene knockdown/knockout, BAC recombineering, inducible gene expression, and HiSeq sequencing, luciferase assays, and electrophysiology.

However, there are limitations to this manuscript.

1. This manuscript is not well organized and is hard to follow. For instance, in the abstract and introduction section, no words on HTT/CHAF1B gene and chromosome 21, but they had done lots of work on these genes (pages 12-14). Figure 1d first appeared on Page 13, almost at the end of the results. This Figure should be moved to Figure S12.
2. The material and methods section is too complicated, and needs to be re-organized, for instance, the generation of 3 transgenic mice needs put together.
3. The result of the function of ZNF91 seems to contrast with previous observations in other cells such as human embryonic stem cells (Ref 18, 19). If ZNF91 is a repressor of SVAs, and SVAs repress the expression of CDK5RAP2, deleting ZNF91 should result in decreasing the expression of CDK5RAP2. But in Figure 2E, deleting ZNF91 actually enhanced the expression of CDK5RAP2. This obviously suggests that SVAs are an enhancer, but not repressors of CDK5RAP2. However, luciferase assay and SVA-F knockdown experiment indicated it is a repressor of CDK5RAP2 (Figure 2, Figure 3). Does this mean ZNF91 is actually an activator of SVAs in these neuronal genes in the NTera-2 cells?
4. In general, SVAs are enhancers of nearby genes (ref 19), but obviously they are repressors of CDK5RAP2 and SCN8A genes in the NTera-2 cells in this study. How to explain this discrepancy?
5. "ZNF91 binds and represses SVA-containing sequences^{17,19}, suggesting SVA-lncRNA AK057321 and ZNF91 might have opposing transcriptional regulatory effects"(page 7, line 3-7) not make sense, as both ZNF91 and AK057321 can suppress SVAs and thus should have same transcriptional regulatory effects.
6. "Moreover, SVA-lncRNA AK057321 counteracted the further repression of HTT SVA produced by ZNF91 (Supplemental Fig. 8e; not examined for CHAF1B SVA). These results indicated SVA sequences and the ZNF91 transcription factor repress gene expression in vitro as previously reported and SVA-lncRNA AK057321 antagonizes these effects" (page 7, lines 12-16) not make sense.
7. In the Luc assay (pages 6-7), both CHAF1B and HTT SVAs should be tested.
8. "The results indicate increasing SVA-lncRNA AK057321 or depleting SVA-repressive ZNF91 drives neuronal differentiation while increasing CDK5RAP2 expression" (page 9, lines 5-7) obviously indicates that ZNF91 is not repressive, but active to SVAs, contrast with previous studies (ref 18-19).
9. If ZNF91 can bind both AK057321 and these genes with SVAs, the effects of AK057321 on these SVAs-containing genes may be regulated by ZNF91 or other KZNF proteins.
10. While this paper proposed another layer of regulation of SVAs, in addition to KZNF proteins, it had not addressed if there is any epigenetic modification on the genes, such as histone marks H3K27ac and H3K4me1 in NTera-2 cells.

Reviewer #2 (Remarks to the Author):

Nadler et al. in this manuscript show that long non-coding RNA AK057321 play an up-regulatory role in neurodevelopment, manifested by de-inhibiting the expression of CDK5RAP2 and SCN8A, and other genes. Both CDK5RAP2 and SCN8A are involved in neuronal differentiation. The potential mechanisms underlying AK057321 are by way of the interactions between SVA (contained within AK057321) and intronic SVAs (of the targeted genes) or ZNF91 (SVA-binding protein). Of note, all the SVA-related regulations/regulators focused in this work are human/primate-specific and disease-related.

The work is comprehensive and the data are intensive, covering from bioinformatics of genomic/patient data all the way to cellular effects and molecular events. The results from this work bridge multiple gaps existing in related research fields. Some key lines of evidence are not de novo, e.g., the hits of SVA-regulatory genes CDK5RAP2 and SCN81 were reported in earlier bioinformatics work (ref. 15); and CDK5RAP2 is known to play an important role in neurogenesis (P17-18 in Discussion). However, the experimental linkage of AK057321 to CDK5RAP2 and SCN8A is initiated from this work, as one major novel aspect of this work.

Overall, the data in the manuscript are generally convincing, incorporating multiple lines of evidence by employing a variety of assays and methods. However, there are a few important viewpoints/concerns that need to be addressed.

1. The electrophysiology regarding sodium channel currents (Fig.2, Fig. 3 and Fig. 4)

The quality of the current traces is low. The amplitude of the inward current is only about tens of pA, in the range that would often raise concerns for whole-cell current recording due to noise issues. Based on the fact that action potentials were elicited normally and the potassium current was in the range of nA, it is suspected (according to H-H ionic mechanisms) that the amplitude of the TTX-sensitive sodium currents should be also in the similar order of magnitude to that of outward potassium currents. Clarification on this matter (with additional experiments) is needed.

2. The actual role of sodium channels

This manuscript gives the impression that CDK5RAP2 and SCN8A are equally important in neuronal development and differentiation. However, in fact, according to this work and literature, CDK5RAP2 is known to be closely involved in the mechanisms of differentiation; in contrast, SCN8A is more like a correlative/collateral marker, which may not mechanistically link to neurogenesis. Because of these limitations, some commonly-used immunocytochemical markers would serve as better indicators of neuronal maturation. Usually sodium channels indirectly participate in development, e.g., through calcium channels and calcium signaling. Therefore, it might be beneficial to examine calcium and calcium-related signals, as another option for data consolidations.

Some minor points:

1. For neuromorphology (e.g., Fig.2), Sholl analysis is expected to help strengthen the conclusion.
2. "Differentiation, maturation, and development" seem interchangeable in the manuscript. Some clarifications or precise selection of wording would be helpful and necessary.

This work, after the above matters are resolved, should provide an important set of information and knowledge, potentially attracting the attention and interests from a broad spectrum of researchers working on neurological diseases, developmental neurobiology, ion channels, gene transcription and regulation, etc.

Reviewer #3 (Remarks to the Author):

The primary concerns are centered around the rationale for studying specific genes and experimental design, these are expanded below. This makes the study somewhat hard to follow and the goals and primary findings unclear. Overall, this manuscript could be streamlined to present a single cohesive study. For instance the title and abstract suggest the manuscript this study focuses on CDK5RAP2/SCN8A control by the SVA-lncRNA AK057321, but then there is additional data on genome-wide observations, CHAF1B, HTT, human chromosome 21 genes, with no clear rationale on why these latter genes were studied, nor what the conclusions are or how they fit into the goals of the study. Moreover, the deletion of SVAs within some of these genes also leads to changes in gene expression suggesting autoregulation of gene expression vis SVAs independent of AK057321. The reader is left with a lot of good data, but no clear idea of the conclusions or findings. Either the manuscript needs to be broadened and the goals of the study stated in the introduction/title, or streamlined and split into multiple papers based on specific themes

Some examples of unclear rationale are highlighted below:

1. What was the rationale behind highlighting the gene ontology terms in Figure 1B? From Sup table 1 there are indeed a number of other processes that have an even greater fold difference (e.g. Hyperbilirubinemia, Distal amyotrophy, Postnatal growth retardation are all higher) that are not highlighted in the figure, nor the text. Perhaps this is because neurodevelopment is the focus of the manuscript, but it should be acknowledged that other disease and biological processes are also enriched to avoid misleading or overrepresenting the sole role in neurodevelopment. The same is true of biological processes where the highlighted enriched processes rank much lower in the list than others (e.g. nervous system development ranks 275th in this list)
2. What was the rationale for selecting the genes for expression analysis with qRT-PCR after AK057321 shRNA-guided knockdown? Especially as RNAseq was then performed in the same experimental paradigm? Why were the results of the RNAseq not just presented given this is a more unbiased and more comprehensive analysis? Moreover, the results for one of these two genes are conflicting. CDK5RAP2 and SCN8A have reduced expression by qRT-PCR, but RNAseq (Sup table 5) shows a non-significant ($p\text{-adj}=0,19$) incremental increase ($\log_2FC=0.12$), for SCN8A the direction is the same and significant. These genes were not assessed in the CRISPR-Cas9 mediated deletion. It's not clear why both shRNA and CRISPR-Cas9 mediated deletion were performed.
3. What is the rationale for studying CHAF1B and HTT by luciferase assay?

Reviewers' comments in black and author response in blue below:

Reviewer #1 (Remarks to the Author):

In humans, SINE-VNTR-Alu (SVA) elements are one of three still active transposable element (TE) families. Using pluripotent human teratoma-derived cell line NTera-2 and three transgenic mice (*HTT*, AK057321 and *CHAF1B*), Nadler et al. found that SVA-lncRNA AK057321 can bind SVAs in the introns of several neuronal genes, such as microcephaly gene *CDK5RAP2* and infantile epileptic encephalopathy gene *SCN8A*. SVAs suppress the expression of these genes, while AK057321 promote their expression. AK057321 knockdown by shRNA or Cas9-sgRNA, or over-expression experiments also shown that it suppresses pluripotency genes (e.g., *NANOG* and *POU5F1*).

The author suggested this AK057321/SVAs network is important for neurodevelopment and neoteny of the hominid brain. Overall, this is a comprehensive and interesting study, that employed multiple advanced technologies including transgenic mice, iPSC, gene knockdown/knockout, BAC recombineering, inducible gene expression, and HiSeq sequencing, luciferase assays, and electrophysiology.

However, there are limitations to this manuscript.

We are pleased reviewer #1 finds our manuscript “is a comprehensive and interesting study, that employed multiple advanced technologies”. We thank this reviewer for their thoughtful comments. Our specific responses to their critiques are addressed below.

1. This manuscript is not well organized and is hard to follow.

For instance, in the abstract and introduction section, no words on *HTT/CHAF1B* gene and chromosome 21, but they had done lots of work on these genes (pages 12-14).

We have added the following sentence to the abstract:

“SVA-lncRNA *AK057321* also upregulates human genes with intronic SVAs (e.g., *HTT*, *CHAF1B* and *KCNJ6*) in mouse brain reconstituting human-specific cortex and cerebellum-enriched expression of *HTT*.”

Figure 1d first appeared on Page 13, almost at the end of the results. This Figure should be moved to Figure S12.

We moved this panel to Supplementary Fig. 2 and refer to it in the sentence below.

“SVA-lncRNA *AK057321* expression is strongest in brain and testis (Supplementary Fig. 2; enriched in cortex and cerebellum, see also the *HTT* gene regulatory studies below, Fig. 5; see also expression of NONHSAT119982.2 in www.noncode.org).”

2. The material and methods section is too complicated, and needs to be re-organized, for instance, the generation of 3 transgenic mice needs put together.

We re-organized Materials and methods section, including combining transgenic mice into one section. For better readability, we moved the details of primer sequences and cloning strategies to the Supplementary Methods Section.

3. The result of the function of ZNF91 seems to contrast with previous observations in other cells such as human embryonic stem cells (Ref 18, 19).

If ZNF91 is a repressor of SVAs, and SVAs repress the expression of *CDK5RAP2*, deleting ZNF91 should result in decreasing the expression of *CDK5RAP2*. But in Figure 2E, deleting ZNF91 actually enhanced the expression of *CDK5RAP2*. This obviously suggests that SVAs are an enhancer, but not repressors of *CDK5RAP2*. However, luciferase assay and SVA-F knockdown experiment indicated it is a repressor of *CDK5RAP2* (Figure 2, Figure 3). Does this mean ZNF91 is actually an activator of SVAs in these neuronal genes in the NTera-2 cells?

Previous studies (e.g., Jacobs et al. 2014, Robbez-Masson et al. 2018 and Haring et al. 2021) have shown that ZNF91 represses expression of genes with an SVA. SVA sequences have binding sites for transcription factors such ZNF91 (Haring et al. 2021) which recruit co-repressors to the SVA sequence (e.g., KAP1 complex, Nielsen et al. 1999 and Sripathy et al. 2006 or the HUSH complex, Robbez-Masson et al. 2018). Our data are consistent with this literature, and we include diagrams of the regulatory pathways in Figs. 2a and 8 for clarity.

In our study, deleting ZNF91 in NTera-2 cells increased expression of *CDK5RAP2* that contains an intronic SVA (Fig. 2c). Deleting the intronic tandem SVAs in *CDK5RAP2* had the same effect as that of deleting ZNF91 – each intervention increased *CDK5RAP2* expression (Figs. 2c and 3c). These data indicate the intronic SVA in *CDK5RAP2* and ZNF91 represses *CDK5RAP2* gene expression.

We have now added additional data to further establish this regulatory mechanism:

- We show CRISPR/Cas9 deletion of *ZNF91* up-regulates expression of several SVA-containing genes: *CDK5RAP2* (Fig. 2c), *SCN8A* (Supplementary Fig. 10c) and *CHAF1B* (Supplementary Fig. 11b, upper graph) in NTera-2 cells.
- We show in the minimal promoter luciferase reporter system that ZNF91 represses luciferase expression when applied to a reporter carrying the SVA from *HTT* (Supplementary Fig. 15e) or from *CHAF1B* (Supplementary Fig. 15f).
- We show deleting the VNTR sequence in the *CHAF1B* SVA (*CHAF1B del-VNTR*), blocks the transcriptional repression cause by ZNF91. This provides further evidence that specific parts of the SVA sequence are required for ZNF91-mediated repression of a gene containing an SVA (Supplementary Fig. 15f).

In summary, it is not that SVAs themselves repress the expression of *CDK5RAP2* and then ZNF91 negates this repression. Rather, it is ZNF91 transcription factor that binds to the SVAs (recruited repressor complexes; as shown in Haring et al. 2021) that represses gene expression.

4. In general, SVAs are enhancers of nearby genes (ref 19; Haring et al.), but obviously they are repressors of *CDK5RAP2* and *SCN8A* genes in the NTERA-2 cells in this study. How to explain this discrepancy?

The transcriptional repressive effects of intronic SVAs found uniformly in our study is likely due to expression of the KRAB-domain family ZNF91 transcription factor in the human NTERA-2 cells we are studying (Fig 1g, red bar; Fig 2c) as shown previously by Robbez-Masson et al. 2018 (Fig 1b). We show ZNF91 represses genes such as *CDK5RAP2* (Fig. 2c), *SCN8A* (Supplementary Fig. 10c) and *CHAF1B* (Supplementary Fig. 11b). Loss of this repressive effect of ZNF91 (Cas9/sgRNA deletion of ZNF91) promotes neuronal maturation (Fig. 2c-i) mimicking effects of overexpressing SVA-lncRNA *AK057321*. Loss of ZNF91 mimics the effect of directly deleting the intronic SVAs in *CDK5RAP2*, both upregulating its expression (Fig. 3c) and promoting neuronal maturation (Fig. 3f-i).

ZNF91 has been shown to repress gene expression by binding to SVAs (Jacobs et al. 2014; Robbez-Masson et al. 2018, Haring et al. 2021).

5. “ZNF91 binds and represses SVA-containing sequences^{17,19}, suggesting SVA-lncRNA *AK057321* and ZNF91 might have opposing transcriptional regulatory effects”(page 7, line 3-7) not make sense, as both ZNF91 and *AK057321* can suppress SVAs and thus should have same transcriptional regulatory effects.

Our data do not indicate that both ZNF91 and SVA-lncRNA *AK057321* suppress SVAs. Rather, our data show ZNF91 represses SVAs and SVA-lncRNA *AK057321* opposes this effect, de-repressing SVAs, allowing neuronal maturation to proceed. See the biological pathway diagrams in Figs. 2a and 8 that illustrate these findings to make it more clear for the reader.

The opposing effects of SVA-lncRNA *AK057321* and ZNF91 are seen in the data described below:

- i. over-expressing *AK057321* SVA-lncRNA upregulates *CDK5RAP2* (Fig. 2b)
- ii. deleting *ZNF91* up-regulates *CDK5RAP2* (Fig. 2c)
- iii. deleting the intronic SVA in *CDK5RAP2* upregulates *CDK5RAP2* (Fig. 3c)

Thus, both the intronic SVA and ZNF91 (which recruits transcriptional repressor complexes to genes by binding to their intronic SVA) repress *CDK5RAP2* expression and this effect is reversed by SVA-lncRNA *AK057321*.

SVA-lncRNA *AK057321* promotes neuronal maturation and this effect is blocked by *CDK5RAP2* shRNA, mimicked by *CDK5RAP2* overexpression, and mimicked by *CDK5RAP2* SVA deletion (Fig. 3).

We show the SVA-lncRNA encoded by *AK057321* forms a complex with the SVAs in *CDK5RAP2* and in *SCN8A* recognized by RNA:DNA heteroduplex antibodies (Fig. 7f). We also

show SVA-lncRNA AK057321 binds ZNF91 (Fig. 7h-i), suggesting a possible decoy mechanism of derepression.

All of this data in combination argues in support of the diagrams shown in the manuscript Figs. 2a and 8 (Fig. 2a shown again below).

6. “Moreover, SVA-lncRNA AK057321 counteracted the further repression of HTT SVA produced by ZNF91 (Supplemental Fig. 8e; not examined for CHAF1B SVA). These results indicated SVA sequences and the ZNF91 transcription factor repress gene expression *in vitro* as previously reported and SVA-lncRNA AK057321 antagonizes these effects” (page 7, lines 12-16) not make sense.

The *in vivo* evidence that SVA-lncRNA AK057321 has opposing effects to ZNF91 are described above in their effects on *CDK5RAP2* and neuronal maturation. The following *in vitro* luciferase reporter assay studies show that SVA-lncRNA AK057321 antagonizes the transcriptional repression produced by an SVA with added ZNF91:

- i. The SVA from *CHAF1B* represses luciferase reporter expression and this repression is partially reversed by SVA-lncRNA AK057321 (Supplementary Fig 15d)
- ii. The SVA from *HTT* represses luciferase reporter expression and this repression is partially reversed by SVA-lncRNA AK057321 (Supplementary Fig. 15e)
- iii. Adding ZNF91 further represses luciferase reporter expression already repressed by the the SVA from HTT and SVA-lncRNA AK057321 de-represses this ZNF91/SVA combination (Supplementary Fig. 15e)

7. In the Luc assay (pages 6-7), both CHAF1B and HTT SVAs should be tested.

1. We have added more genes to show the transcriptional repressive effects of intronic SVAs (including *CHAF1B*, *HTT*, *CDK5RAP2* and others; Supplementary Fig. 15).
2. We demonstrate that deleting the VNTR from the *CHAF1B* SVA blocks both the transcriptional repression produced by the SVA and the additional repression produced by adding ZNF91.
3. Finally, for both *CHAF1B* and *HTT* SVAs, we show SVA-lncRNA *AK057321* partially reverses the repression caused by these two SVAs on the luciferase reporter (Supplementary Fig. 15).

8. “The results indicate increasing SVA-lncRNA *AK057321* or depleting SVA-repressive ZNF91 drives neuronal differentiation while increasing *CDK5RAP2* expression” (page 9, lines 5-7) obviously indicates that ZNF91 is not repressive, but active to SVAs, contrast with previous studies (ref 18-19).

Whether an SVA is repressive or rather enhancing of gene expression may depend on the cell context. In NTERA-2 cells, likely due to the expression of ZNF91 and possibly other gene regulatory factors, SVAs repress all of the genes examined. Deleting the SVA in *CDK5RAP2* (Fig. 3c) or in *CHAF1B* (Supplementary Fig. 11d) increased their expression levels. Further support is the finding that “depleting ZNF91 increases *CDK5RAP2* expression while initiating neuronal maturation”.

”.

- These results indicate ZNF91 represses the gene *CDK5RAP2* (Fig. 2c) and inhibits neuronal maturation (Fig. 2d-i). This likely occurs through the SVA in *CDK5RAP2* because deleting that SVA both drives neuronal maturation (Figs. 3f-i) and increases *CDK5RAP2* expression (Fig. 3c). The results indicate the intronic SVA in *CDK5RAP2* blocks neuronal maturation. Importantly, knock-down of *CDK5RAP2* with an shRNA negates the ability of SVA-lncRNA *AK057321* to promote neuronal maturation (Figs. 3d and 3f-i), indicating de-repression of *CDK5RAP2* by SVA-lncRNA *AK057321* is required.

9. If ZNF91 can bind both *AK057321* and these genes with SVAs, the effects of *AK057321* on these SVAs-containing genes may be regulated by ZNF91 or other KZNF proteins.

- We do find ZNF91 binds SVA-lncRNA *AK057321* (Fig. 7h-i).
- See Fig 2a and Fig. 8 for the biological pathway diagrams supported by our data where ZNF91 and *AK057321* interact to reciprocally regulate genes with an intronic SVA.
- We agree that other KRAB-domain ZNFs (KZNF) could potentially play a role in regulating some genes with an intronic SVA and some biological processes that this gene regulation controls. But here we show that deleting ZNF91 is sufficient to reconstitute the gene regulatory and neuronal maturation promoting effects that are the focus of this current study (Fig. 2c, d-i).

10. While this paper proposed another layer of regulation of SVAs, in addition to KZNF proteins, it had not addressed if there is any epigenetic modification on the genes, such as histone marks H3K27ac and H3K4me1 in NTERA-2 cells.

- This study uncovers and focuses on a major biological function of this SVA gene regulatory mechanism, the delay of neuronal maturation which is characteristic of hominids and most prominent in human. We have implicated specific genes in the biological process, *CDK5RAP2* and *SCN8A* through their human-specific SVAs.
- The study already has extensive experimental work and data and from that data is able to make major conclusions. To make a study of the associated epigenetic modification meaningful one would need to further investigate the molecular mechanisms of those epigenetic changes and determine which are important to the changes in gene expression and biological functions we have described. This would require in depth analysis of the protein complexes associated with SVA-lncRNA AK057321 and ZNF91 in the NTerra2 cell system followed by interventions on the discovered molecular species. This is not feasible in a reasonable time frame and would be better served in a separate manuscript.

Reviewer #2 (Remarks to the Author):

Nadler et al. in this manuscript show that long non-coding RNA AK057321 play an up-regulatory role in neurodevelopment, manifested by de-inhibiting the expression of *CDK5RAP2* and *SCN8A*, and other genes. Both *CDK5RAP2* and *SCN8A* are involved in neuronal differentiation. The potential mechanisms underlying AK057321 are by way of the interactions between SVA (contained within AK057321) and intronic SVAs (of the targeted genes) or ZNF91 (SVA-binding protein). Of note, all the SVA-related regulations/regulators focused in this work are human/primate-specific and disease-related. The work is comprehensive and the data are intensive, covering from bioinformatics of genomic/patient data all the way to cellular effects and molecular events. The results from this work bridge multiple gaps existing in related research fields. Some key lines of evidence are not de novo, e.g., the hits of SVA-regulatory genes *CDK5RAP2* and *SCN8A* were reported in earlier bioinformatics work (ref. 15); and *CDK5RAP2* is known to play an important role in neurogenesis (P17-18 in Discussion). However, the experimental linkage of AK057321 to *CDK5RAP2* and *SCN8A* is initiated from this work, as one major novel aspect of this work. Overall, the data in the manuscript are generally convincing, incorporating multiple lines of evidence by employing a variety of assays and methods. However, there are a few important viewpoints/concerns that need to be addressed.

We appreciate that reviewer #2 finds our “work is comprehensive and the data are intensive, covering from bioinformatics of genomic/patient data all the way to cellular effects and molecular events”. We agree, “this work bridge multiple gaps existing in related research fields” and “the experimental linkage of *AK057321* to *CDK5RAP2* and *SCN8A* is initiated from this work, as one major novel aspect of this work”. Thank you for your comment below that this will “potentially attract the attention and interest of a broad spectrum of researchers working on neurological diseases, developmental neurobiology, ion channels, gene transcription and regulation, etc.” We address the reviewer’s concerns below.

1. The electrophysiology regarding sodium channel currents (Fig.2, Fig. 3 and Fig. 4)

The quality of the current traces is low. The amplitude of the inward current is only about tens of pA, in the range that would often raise concerns for whole-cell current recording due to noise issues. Based on the fact that action potentials were elicited normally and the potassium current was in the range of nA, it is suspected (according to H-H ionic mechanisms) that the amplitude of the TTX-sensitive sodium currents should be also in the similar order of magnitude to that of outward potassium currents. Clarification on this matter (with additional experiments) is needed.

The goal of our study was to test the ability of SVA-lncRNA *AK057321* and other interventions to initiate neuronal maturation (see Fig. 4). Importantly, we show action potentials are normal in size only in fully matured Ntera-2 cells driven to mature into glutamatergic neurons using transcription factor *Ngn2* co-cultured with astrocytes for 6-weeks (see Fig 1d). Action potentials are small in all of our other studies 5-15 mV (Fig. 4f-g, see also traces in Supplementary Fig 10d) because we performed measurements after 15-days or less of maturation (in absence of *Ngn2* that rapidly accelerates maturation).

Additional detailed comments below:

- Consistent with the attenuated sodium spikes in these 15-day or less studies, the sodium currents are expected to be small.
- The overall steady state potassium currents are only ~600 pA (Supplementary Fig. 8). This is a mixture of potassium currents including some that control resting membrane potential which has become hyperpolarized with maturation in parallel with the increased potassium currents. We do not investigate the specific genes contributing to these potassium currents and therefore moved this data to Supplementary Figs.
- We test and compare the ability of SVA-lncRNA *AK057321* and other interventions to drive early neuronal maturation which does not otherwise occur in the absence of these interventions (see Fig. 4g).

2. The actual role of sodium channels

This manuscript gives the impression that *CDK5RAP2* and *SCN8A* are equally important in neuronal development and differentiation. However, in fact, according to this work and literature, *CDK5RAP2* is known to be closely involved in the mechanisms of differentiation; in contrast, *SCN8A* is more like a correlative/collateral marker, which may not mechanistically link to neurogenesis. Because of these limitations, some commonly-used immunocytochemical markers would serve as better indicators of neuronal maturation. Usually sodium channels indirectly participate in development, e.g., through calcium channels and calcium signaling. Therefore, it might be beneficial to examine calcium and calcium-related signals, as another option for data consolidations.

We agree that *CDK5RAP2* is more central to the process of progenitor to neuron maturation. We describe the findings for *SCN8A* as an example of many other genes containing an SVA which may contribute to specialized aspects of neuronal maturation. *SCN8A* function is readily assayed and it is a major contributor to the voltage-gated sodium currents during early neuronal maturation as shown by our findings.

A key feature of neuronal maturation and matured neuronal function is the generation of voltage-gated sodium currents and action potentials. Studies of *SCN8A* are important in this initial characterization because of the human-specific SVA and its known major role in contributing to the earliest sodium channel currents during neuronal maturation. This early functional role is further highlighted by its known role in human genetic disorders of early infantile epilepsy and intellectual disability.

SCN8A is not solely a correlative biomarker, but instead has a human-specific SVA and illustrates how intronic SVAs regulate multiple specific aspects of neuronal maturation. This is an important conclusion which supports the idea that intronic SVAs might regulate neuronal maturation at multiple levels (see genes with intronic SVA regulated by SVA-lncRNA Ak057321 underlying a diversity of important neuronal functions in Fig. 1g and mentioned in the discussion)

“commonly-used immunocytochemical markers would serve as better indicators of neuronal maturation”:

To compare different molecular-genomic interventions, we needed quantitative measures afforded by whole cell patch clamp electrophysiology (resting membrane potential, ionic currents, capacitance ~size, and sodium spikes) and neurite outgrowth (now including Sholl analysis). These are well established ways to achieve a quantitative measure of neuron functional maturation and revealed differences in maturation of iPSC-derived neurons from human and chimpanzee (Marchetto et al. 2019; <https://doi.org/10.7554/eLife.37527>).

Immunohistochemical stains give only semi-quantitative measures.

qRT-PCR, performed in our study, has already provided an alternative quantitative measure of neuronal maturational and pluripotency molecular markers.

“it might be beneficial to examine calcium and calcium-related signals, as another option for data consolidations”:

While potentially of interest for future study we did not see an SVA in a major calcium channel forming genes, but rather only in an accessory protein (SVA in *CACNA2D4*, Supplementary Fig. 7). As can be seen from the large number of neuronal genes containing intronic SVAs (Supplementary Figs. 4-7), there will be many future studies to pursue in follow up to this first report.

Some minor points:

1. For neuromorphology (e.g., Fig.2), Sholl analysis is expected to help strengthen the conclusion.

Thank you for the suggestion, this method is now included and substantially improves the quality of our quantitative analysis of neuronal morphologic maturation (Fig. 2l-p and Supplementary Fig. 9)

2. "Differentiation, maturation, and development" seem interchangeable in the manuscript. Some clarifications or precise selection of wording would be helpful and necessary. This work, after the above matters are resolved, should provide an important set of information and knowledge, potentially attracting the attention and interests from a broad spectrum of researchers working on neurological diseases, developmental neurobiology, ion channels, gene transcription and regulation, etc.

To be consistent, we now use neuronal maturation throughout the article.

Reviewer #3 (Remarks to the Author):

The primary concerns are centered around the rationale for studying specific genes and experimental design, these are expanded below. This makes the study somewhat hard to follow and the goals and primary findings unclear. Overall, this manuscript could be streamlined to present a single cohesive study. For instance the title and abstract suggest the manuscript this study focuses on CDK5RAP2/SCN8A control by the SVA-lncRNA AK057321, but then there is additional data on genome-wide observations, CHAF1 B, HTT, human chromosome 21 genes, with no clear rationale on why these latter genes were studied, nor what the conclusions are or how they fit into the goals of the study.

We appreciate this reviewer's concerns regarding cohesion and readability of our study. We have modified the manuscript and figures to achieve a better flow of the data (neuronal maturation, neuronal gene expression, and gene regulatory mechanisms) used to provide compelling evidence that SVA-lncRNA, ZNF91, and intronic SVAs interact to confer species-specific gene regulation and control an important property, neuronal maturation. We made changes listed below:

1. Our study focuses on identifying molecular mechanisms that might explain human-specific features such as delayed neuronal maturation. We provide data suggesting the hominid specific retrotransposon SVA, when present in a gene intron, act to repress expression of the gene and being present in key neurodevelopmental and neuronal functional genes act to delay neuronal maturation. We show SVA-lncRNA *AK057321* expression increases as progenitor cells mature into neurons and that it is necessary to increase SVA-lncRNA *AK057321*, remove ZNF91-mediated transcriptional repression, or delete the intronic SVA to initiate neuronal maturation.
 - We focus on SVA-lncRNA *AK057321*'s role in progenitor to neuron maturation by moving the evidence that SVA-lncRNA *AK057321* is up-regulated during neuronal maturation to the first figure (Figs. 1c-d).
 - We added a new biological pathway diagrams (Figs. 2a and 8) to help the reader understand how the data are interpreted.

- We also changed the title to emphasize this study focuses on SVA-lncRNA *AK057321*'s role in progenitor to neuron maturation.
- 2. However, we also think it is important to share our evidence that this gene regulatory mechanism is not limited to a cell culture system but also occurs in brain *in vivo*. The finding in human brain that *HTT* gene expression is enriched in cortex and cerebellum relative to brainstem and that this is reconstituted in mouse brain when transgenic human *HTT* and human SVA-lncRNA *AK057321* are combined translates the findings back to the human brain.
 - We have streamlined the abstract and now include our findings on human *HTT* and human chromosome 21 genes, as requested.
- 3. Regarding our choice of genes to focus on in this study:

CDK5RAP2:

- *CDK5RAP2* has major role in human microcephaly is required during neurogenesis²²⁻²⁸
- *CDK5RAP2* has an intronic human-specific tandem SVA_F (Fig. 3a)
- *CDK5RAP2* is strongly downregulated with *AK057321* loss (Fig. 1g-i)
- *CDK5RAP2* is upregulated by multiple interventions that promote neuronal maturation such as increased *Ngn2* (Fig. 1c) or SVA-lncRNA *AK057321* (Fig. 2b) or decreased *ZNF91* (Fig. 2c) expression].

SCN8A:

- Mutations in *SCN8A* to cause early infantile epileptic encephalopathy and cognitive impairment³⁰⁻³².
- Increasing and decreasing SVA-lncRNA *AK057321* upregulates and down-regulates, respectively, *SCN8A* expression (Supplementary Fig. 10a-b, Fig. 1g).
- CRISPR/Cas9-sgRNA mediated deletion of the SVA-repressive *ZNF91* upregulates *SCN8A* expression (Supplementary Fig. 10c).

As we suggest in the Discussion section, genes with intronic SVAs are enriched for neurodevelopmental and neurodevelopmental disease genes (Supplementary Data 3-4) with a diversity of functions. The contributions of the human-specific SVAs within *CDK5RAP2* and *SCN8A* serve as founding examples of intragenic SVAs that have distinct roles in the maturation of human neuronal functions.

- 4. Finally, we provide foundational data to explain the molecular mechanisms whereby this regulation occurs targeting intervention with a variety of SVAs in *in vitro* and *in vivo* systems (Fig. 7, Supplementary Fig. 15). We include some genes with intronic SVAs out of convenience of having mice carrying human full-length genes (e.g., *HTT* or Chromosome 21-*CHAF1B*).

Moreover, the deletion of SVAs within some of these genes also leads to changes in gene expression suggesting autoregulation of gene expression via SVAs independent of AK057321.

ZNF91, expressed in NTera-2 cells (Fig. 1g, red bar), has been shown to repress gene expression through binding to SVAs (Jacobs et al. 2014; Robbez-Masson et al. 2018). SVA sequences are binding sites for transcription factors such as ZNF91 (Haring et al. 2021) which recruit co-repressors to the SVA sequence (e.g., KAP1 complex, Nielsen et al. 1999 and Sripathy et al. 2006 or the HUSH complex, Robbez-Masson et al. 2018). When deleting the SVA in NTera-2 cells, we would be removing the ZNF91 target sequence and thereby releasing the gene from ZNF91-mediated gene repression (see biological pathway diagrams in Figs 2a and 8).

- Deleting the *ZNF91* gene in human embryonic stem cells was shown to increase expression of numerous genes located nearby SVA sequences¹⁴, confirming it represses gene expression *in vivo* through its binding to SVAs.
- Our data show deleting the *ZNF91* gene in NTera-2 cells up-regulates *CDK5RAP2* expression and promotes neuronal maturation (Fig. 2).
- Our data are consistent with this literature and we include diagrams (Figs. 2a and 8) to illustrate these mechanisms for the reader.

See similar comments above, reviewer #2.

The reader is left with a lot of good data, but no clear idea of the conclusions or findings.

- The slowing of neuronal maturation in human compared to other hominid species is the motivation of this study and is now more clearly stated in the abstract and introduction.
- The added biological pathway diagrams (Figs. 2a and 8) now illustrate the functions and mechanisms under study in this manuscript.
- We explain that the intronic SVA and the repressive transcription factor that binds SVAs reduce expression of genes such as *CDK5RAP2* and *SCN8A* which has the effect of delaying neuronal maturation. Expression of SVA-lncRNA *AK057321*, which occurs as progenitors are matured in neurons, releases the intronic SVA-ZNF91 gene repression to allow neuronal maturation to proceed.
- We have improved the writing throughout the manuscript.

Either the manuscript needs to be broadened and the goals of the study stated in the introduction/title, or streamlined and split into multiple papers based on specific themes.

Thank you for the suggestion. We have broadened the conclusions in the abstract and introduction. We are open to changing the title to a broader statement such as the running title. We placed the strongest aspects of the story and specific genes in the title.

While one might consider splitting this manuscript into multiple smaller stories, we prefer a single rigorous study all packaged together rather than having the results spread across multiple papers. These data provide a foundation for us and others to further expand on any part of the manuscript. We prefer the data go into the public domain and that it be connected to these other elements of the story.

Some examples of unclear rationale are highlighted below:

1. What was the rationale behind highlighting the gene ontology terms in Figure 1 B? From Supplementary Table 1 there are indeed a number of other processes that have an even greater fold difference (e.g. Hyperbilirubinemia, Distal amyotrophy, Postnatal growth retardation are all higher) that are not highlighted in the figure, nor the text. Perhaps this is because neurodevelopment is the focus of the manuscript, but it should be acknowledged that other disease and biological processes are also enriched to avoid misleading or overrepresenting the sole role in neurodevelopment. The same is true of biological processes where the highlighted enriched processes rank much lower in the list than others (e.g. nervous system development ranks 275th in this list).

We have modified the manuscript to account for other GO term enrichments as follows:

“Using gene ontology term enrichment analysis, we found that genes containing intronic SVAs are enriched for those involved in neurodevelopment and neurodevelopmental disease (Fig. 1f, Supplementary Data 3-4; enrichment of gene ontology terms for other biological functions and diseases were also seen).”

Amongst disease GO terms, intellectual disability was the most significant. Cerebral atrophy, often found associated with intellectual disability, was also high. This motivated the focus on neurodevelopmental genes which underlie intellectual disabilities and microcephaly. When comparing hominid species, one prominent feature is the larger brain and slowed neuronal maturation in human compared to others as we explain in the introduction. This motivated the emphasis on neuronal maturation effects of these intronic SVAs and SVA regulators (SVA-lncRNA *AK057321* and *ZNF91*). We think we have improved the rationale for the study in the updated manuscript.

To understand the potential functional relevance of intronic SVAs (particularly those that are human-specific), we prioritized gene ontology analysis in human disease where we found greatest enrichment for intellectual disability and developmental delay (Fig. 1f). This is why neurodevelopment is the focus of this manuscript. We find similar enrichments in the category of biological functions (Fig. 1f). We framed the study around explaining the delay in neuronal maturation and this is explained by the transcriptional repression by intronic SVAs (via *ZNF91*) in *CDK5RAP2* and *SCN8A* and likely others not yet studied. We agree, there are many other lines of research one could have pursued, but we focused on these neuronal functions.

All the other information is readily available to the reader in supplementary materials.

2. What was the rationale for selecting the genes for expression analysis with qRT-PCR after *AK057321* shRNA-guided knockdown?

Again, amongst disease go terms, intellectual disability was the most significant. Cerebral atrophy, often found in intellectual disability was also high. This motivated the focus on neurodevelopmental genes with intronic SVAs known to underlie human intellectual disabilities and microcephaly.

This approach led us to successfully identify key human-specific intronic SVAs that slow neuronal maturation, the ones in *CDK5RAP2* and *SCN8A*. The others which we highlight in the qRT-PCR are likely to play roles in other neuronal maturational properties which we have not yet measured such as synaptogenesis (*NRXN1*, *NRXN2*, *SHANK2*) or the maturational change of GABAergic synaptic transmission from depolarizing to hyperpolarizing (*WNK3*) as we describe in the discussion of the manuscript.

Especially as RNAseq was then performed in the same experimental paradigm? Why were the results of the RNAseq not just presented given this is a more unbiased and more comprehensive analysis? Moreover, the results for one of these two genes are conflicting. *CDK5RAP2* and *SCN8A* have reduced expression by qRT-PCR, but RNAseq (Sup table 5) shows a non-significant ($p\text{-adj}=0.19$) incremental increase ($\log_2\text{FC}=0.12$), for *SCN8A* the direction is the same and significant.

RNA seq is not a reliable quantitative assay. Also, many of the genes containing SVAs will cause cascading changes in gene expression down-stream. For both of these reasons, we focused on a more quantitative assay, qRT-PCR, to measure the regulatory effects of the SVA-lncRNA and *ZNF91* on genes containing an intronic SVA. Typically, lncRNAs regulate gene expression and the complementary sequence in the SVA-lncRNA and the genomic intronic SVAs suggested possible regulation through complementary annealing (as we found occurs at the end of the manuscript in our mechanism-focused studies).

The RNA seq study was exploratory with a limited N but did support our conclusion that these pathways regulate neuronal maturation where loss of SVA-lncRNA *AK057321* increased pluripotency markers (see confirmation in Fig. 1j) and reduced neuronal differentiation markers. The findings were established using more quantitative methods of qRT-PCR (Fig. 1j), neuronal electrophysiology (Fig. 2d-i), and neuronal morphology (Fig. 2j-p), and so the RNA seq data (being an unreliable source of quantitation) were moved to the Supplementary Note.

These genes (*CDK5RAP2* and *SCN8A*) were not assessed in the CRISPR-Cas9 mediated deletion. Its not clear why both shRNA and CRISPR-Cas9 mediated deletion were performed.

We have updated Fig 1 to include both shRNA (Fig. 1g) and Cas9/sgRNA (Fig. 1j) methods used to deplete SVA-lncRNA *AK057321* and measure effects on *CDK5RAP2* and *SCN8A* expression. The Cas9/sgRNA results confirm the shRNA results, further strengthening our conclusions.

What is the rationale for studying *CHAF1B* and *HTT* by luciferase assay?

We now include a wider range of SVAs in the luciferase assay including the gene shown to have major functional effects, *CDK5RAP2* (Supplementary Fig. 15g). We studied *CHAF1B* and *HTT* in the luciferase assay because of our findings that SVA-lncRNA *AK057321* regulates these gene in mice carrying the full-length human genes. Such human gene carrying mice are limited. *CHAF1B* is on human chromosome 21 in Chr21 transchromosomic mice and *HTT* in YAC transgenic mice. We do show *CHAF1B* is important to regulating progenitor cell differentiation and deletion of its SVA increase pluripotency markers and cell proliferation (Supplementary Fig. 11), but this is a complex function that does not align simply into the progenitor to neuron maturation pathway studied in this manuscript.

The luciferase assay has served as a convenient system published by multiple other scientists (Jacob et al. 2014; Imbault et al. 2017; Robbez-Masson et al. 2018) for mechanistic studies of SVA nucleic acid sequence effects on gene expression.

Including multiple SVAs to demonstrate luciferase gene repression further strengthens the conclusions of the manuscript.

REVIEWERS' COMMENTS:

Reviewer #1 (Remarks to the Author):

The authors addressed all my concerns carefully. I suggest it is published in communications biology.

Reviewer #2 (Remarks to the Author):

The authors have addressed most of my concerns. I only have one comment in this round: Regarding the immunocytochemical markers/assays for neuronal maturation, it is still commonly practiced in addition to other approaches, including qPCR (this work) and RNA-Seq (Marchetto et al. 2019 mentioned in the response letter), etc. In Marchetto et al. 2019, staining data were shown for Tuj1 (also in Figure 1 of this work), Nestin, Map2, and Synapsin1. The suggestion in the last round was meant to conduct immunostaining by these conventional antibodies including those targeting some Ca²⁺ signaling molecules (Sarnat 2014, PMID: 25227552). And now as the authors explained, the stage of the neurons was relatively early (15-day or less) which may not show pronounced staining signals. However, since some indices were significant (such as sholl analysis results) and some were not or less confirmative (such as sodium currents), it would be helpful to examine the maturation profiles in the context of common biomarkers. It is worth some discussion if these biomarkers may not be well assessable at this early stage.

Reviewer #3 (Remarks to the Author):

The manuscript is much improved from the previous version.

Overall the authors have decided to include all data in the manuscript, and broadened the abstract to be somewhat more descriptive of the study. I think that the running title should be used instead of the current title as it more accurately describes the data. Moreover, the introduction, while now a little more broad essentially is only an introduction for the first paragraph. The next two paragraphs are a summary of the paper. I think a better more concise introduction that places the work in context will help to convey the main implications of the findings (rather than stating them). this will help with the overall big picture that both reviewer 1 and I emphasized as a limitation of the manuscript. Overall the emphasis should be on AK057321 regulatory role on neurodevelopmental genes and the mechanisms that underpin this, and this needs to be framed in the introduction. This framing of mechanism will allow the authors to introduce ZNF91 and its role, thus aiding interpretability of their results. The authors present a large amount of work, which is interesting and relevant, but this needs to be placed in broad context so that the reader can appreciate the results.

Please provide a brief rebuttal letter that addresses the remaining concerns from Reviewers #2-3, and a marked-up version of the text outlining any relevant edits:

Reviewer #2:

Regarding the immunocytochemical markers/assays for neuronal maturation, it is still commonly practiced in addition to other approaches, including qPCR (this work) and RNA-Seq (Marchetto et al. 2019 mentioned in the response letter), etc. In Marchetto et al. 2019, staining data were shown for Tuj1 (also in Figure 1 of this work), Nestin, Map2, and Synapsin1. The suggestion in the last round was meant to conduct immunostaining by these conventional antibodies including those targeting some Ca²⁺ signaling molecules (Sarnat 2014, PMID: 25227552). And now as the authors explained, the stage of the neurons was relatively early (15-day or less) which may not show pronounced staining signals. However, since some indices were significant (such as shall analysis results) and some were not or less confirmative (such as sodium currents), it would be helpful to examine the maturation profiles in the context of common biomarkers. It is worth some discussion if these biomarkers may not be well assessable at this early stage.

Reviewer #2: We use the phrasing initiate maturation throughout including the title.

-That phrasing helps the reader understand these neurons are not fully mature.

-We disagree with the comment “some were not or less confirmative (such as sodium current).”

1. The voltage-gated sodium currents are robust statistically significant measures (Fig. 2d,e and 4f) that undergo maturational changes (Fig. 4h) and are inhibited by tetrodotoxin (Fig. 2e).
2. -The sodium spikes (generated by voltage-gated sodium channels) are also robust measures undergoing maturational changes in these cells and are inhibited by tetrodotoxin (Fig. 4f-g and Supplementary Fig. 12d-e).
3. -Both measures of the voltage-gated sodium channel currents are influenced by deletions of the intronic SVA within the gene encoding the voltage-gated sodium channel *SCN8A* making a direct connection of the gene and its function (Fig. 4c, f-h).

-We examined the marker Map2 and the staining was weak and was therefore not included.

Reviewer #3:

Moreover, the introduction, while now a little more broad essentially is only an introduction for the first paragraph. The next two paragraphs are a summary of the paper. I think a better more concise introduction that places the work in context will help to convey the main implications of the findings (rather than stating them). this will help with the overall big picture that both reviewer 1 and I emphasized as a limitation of the manuscript. Overall the emphasis should be on AK057321 regulatory role on neurodevelopmental genes and the mechanisms that underpin this, and this needs to be framed in the introduction. This framing of mechanism will allow the authors to introduce ZNF91 and its role, thus aiding interpretability of their results. The authors

present a large amount of work, which is interesting and relevant, but this needs to be placed in broad context so that the reader can appreciate the results.

Reviewer #3: We edited the introduction substantially to help the reader better understand the context and rationale for the study.

As requested by the editor, we trimmed the introduction, results, and discussion to achieve <5000 words. Concurrently, we simplified the language to make it more accessible to a general reader. Highlighting added to areas with substantial editing.